# QPKO: Differentiable QP-Embedded Deep Koopman Framework for Modeling Nonlinear Systems

**Runze Tian** [1]  **Peng Kou** [1]

## Abstract

Deep learning has been widely regarded as a powerful tool for Koopman operator theory-based modeling, as it provides a promising architecture for data-driven learning of observable functions. To fully leverage this advantage, a well-designed training paradigm is required. However, the existing training paradigms typically either incur high optimization complexity or hinder effective end-to-end training, limiting modeling accuracy and training efficiency. To address this issue, we propose a differentiable quadratic programming (QP)-embedded deep Koopman framework (QPKO). In QPKO, a QP problem, which comprises a one-step accuracy-oriented objective function and a set of multi-step accuracy-oriented constraints, is formulated to introduce a mapping from observable functions to the global linear model. By doing so, the global linear model no longer needs to be treated as an independent trainable component, thereby effectively reducing optimization complexity. This QP-based mapping is implemented as a differentiable and computationally efficient module by leveraging OptNet (a differentiable QP layer), enabling effective end-to-end training. Experiments on four nonlinear dynamical systems show that QPKO achieves satisfactory improvements in modeling accuracy, training efficiency, and control performance.

## 1. Introduction

Nonlinear dynamical systems are ubiquitous in many fields, including renewable energy (Wang et al., 2025), robotics (Zhao et al., 2025), and aerospace engineering (Liu et al., 2025). The key to applying the well-established linear control and analysis techniques to these systems lies in constructing their global linear dynamic models (Sakib & Pan, 2025; Hernández-Ortega & Messina, 2018). However, the complex nonlinearity of these systems poses a challenge for achieving this goal.

The Koopman operator theory (Koopman, 1931) provides a mathematical foundation to address this challenge, as it shows that nonlinear dynamics can be described as infinite-dimensional linear dynamics in a space of observable functions. To bring this theory into the practice of linear modeling, numerous data-driven frameworks, referred to as Koopman frameworks, have been proposed. They identify observable functions and approximate the infinite-dimensional linear dynamics with a finite-dimensional global linear model in a lifted space. These frameworks are commonly developed based on dynamical mode decomposition (DMD) (or its variants) (Schmid, 2010; Williams et al., 2015a) and deep learning (Yeung et al., 2019). Compared to the former, the latter is widely considered to have greater potential in modeling accuracy, since it provides a scalable and data-driven architecture for the discovery and representation of observable functions, thereby eliminating the expertise-dependent manual observable design (Lusch et al., 2018).

Nevertheless, to fully leverage the above advantage of deep learning in Koopman modeling, a well-designed training paradigm is still required. Unfortunately, although the training paradigms adopted by the existing deep learning-based Koopman frameworks have shown promising performance, they typically incur high optimization complexity or hinder effective end-to-end training. This limits further improvements in modeling accuracy and training efficiency.

Specifically, the existing deep learning-based Koopman frameworks typically follow two training paradigms. In the first paradigm (Lusch et al., 2018; Jin et al., 2025b), observable functions and the global linear model are parameterized as two independent trainable components. Although this design facilitates end-to-end training, introducing two independent trainable components results in a large search space. This complicates the underlying optimization problem, degrading the modeling accuracy and training efficiency.

In comparison, the second paradigm (Li et al., 2017; Xu

---

[1]School of Electrical Engineering, Xi'an Jiaotong University, Xi'an, China. Correspondence to: Peng Kou <koupeng@mail.xjtu.edu.cn>.

*Proceedings of the 43rd International Conference on Machine Learning*, Seoul, South Korea. PMLR 306, 2026. Copyright 2026 by the author(s).

et al., 2025) does not treat the global linear model as an independent trainable component. Instead, it formulates a one-step accuracy-oriented least-squares (LS) problem, which introduces a mapping from the observable functions to the corresponding global linear model. As a result, only the observable functions need to be optimized during training, which reduces the search space and simplifies the optimization. Nevertheless, incorporating the LS problem often leads to an alternating two-stage training procedure. That is, observable functions and the global linear model are updated in separate stages with the other component kept fixed. This prevents end-to-end back-propagation, thereby hindering stable training convergence and degrading modeling accuracy. A few frameworks (Iwata & Kawahara, 2023) attempt to enable end-to-end training by directly differentiating the closed-form LS solution using automatic differentiation. Unfortunately, since this solution includes pseudoinverse operation, this differentiation process can be numerically unstable and gives fluctuating training gradients. Moreover, in this paradigm, the LS problem only considers one-step prediction accuracy, which is misaligned with the overall training loss that evaluates multi-step rollout error.

Motivated by the above issues, we propose a differentiable quadratic programming (QP)-embedded deep Koopman framework (QPKO). In QPKO, a QP problem is formulated to introduce the mapping from observable functions to the global linear model. The objective function of this QP problem is developed to minimize the one-step prediction error, while a set of infinity norm-based constraints is designed to enhance the multi-step prediction accuracy. Thanks to this QP-based mapping, the global linear model is no longer treated as an independent trainable component. Instead, it is determined by the currently learned observable functions in a convex optimization manner, which accounts for both one-step and multi-step accuracy. This effectively reduces the training complexity and provides a basic assurance for modeling accuracy. Meanwhile, by leveraging OptNet (a differentiable QP layer) (Amos & Kolter, 2017), this QP-based mapping is implemented as a differentiable and computationally efficient module, enabling effective end-to-end training.

## 2. Related Work

**Koopman Frameworks.** Building on Koopman operator theory (Koopman, 1931), numerous Koopman frameworks have been proposed to establish a global linear model for nonlinear dynamical systems. Some studies, especially early ones, adopted DMD (Rowley et al., 2009; Schmid, 2010; Proctor et al., 2016; Kutz et al., 2016) and its variants, such as EDMD (Williams et al., 2015a; Korda & Mezic, 2018; Mauroy & Goncalves, 2020), kernel DMD (Williams et al., 2015b), ResDMD (Colbrook & Townsend, 2024), and Han-

kel DMD (Sakib & Pan, 2025), to develop such frameworks. Since these frameworks rely on manual design of observable functions, their performance heavily depends on domain expertise. Subjective or arbitrary choices of observables can substantially degrade their modeling accuracy.

To address this issue, recent work has placed increasing focus on constructing deep learning-based frameworks, which leverage NNs, such as MLP (Lusch et al., 2018; Yeung et al., 2019; Han et al., 2020; Shi & Meng, 2022; Jin et al., 2025b), GNN (Li et al., 2020), Hierarchical NN (Wang et al., 2024), and KAN-Transformer (Zhao et al., 2025), to parameterize the observable functions. By doing so, observable functions can be optimized during training, eliminating the manual design. These frameworks follow two training paradigms. The first paradigm parameterizes the observable functions and the global linear model as two independent trainable modules and optimizes them simultaneously in an end-to-end manner (Lusch et al., 2018; Yeung et al., 2019; Shi & Meng, 2022; Wang et al., 2024; Jin et al., 2025b; Zhao et al., 2025). The second paradigm generates the global linear model based on observable functions via a one-step accuracy-oriented LS problem. Most frameworks in this paradigm (Li et al., 2017; Han et al., 2020; Xu et al., 2025) carry out an alternating two-stage training, while a few (Iwata & Kawahara, 2023) try to enable an end-to-end training by back-propagating through the LS problem via automatic differentiation.

While these deep learning-based frameworks have achieved promising performance, their training paradigms limit the training efficiency and modeling accuracy. Our QPKO overcomes this limitation by formulating a QP-based mapping to effectively reduce the optimization complexity and leveraging OptNet to achieve numerically stable and computationally efficient end-to-end training.

**Differentiable QP Layer.** Previous studies have developed numerous methods, which enable embedding QP into an end-to-end deep learning framework as a differentiable layer, ranging from QP-specific approaches (Amos & Kolter, 2017; Butler & Kwon, 2023) to general frameworks (Agrawal et al., 2019a; Blondel et al., 2022; Ren et al., 2023; Agrawal et al., 2019b; Pan et al., 2024) applicable to broad classes of convex programs. Among them, OptNet (Amos & Kolter, 2017) is one of the most representative methods. In OptNet, a primal-dual interior point solver is tailored for GPU-based batched QP solving. In back-propagation, OptNet does not directly differentiate through the QP solver using automatic differentiation. Instead, it introduces a KKT-based implicit differentiation method to compute gradients in a numerically stable and nearly "computationally free" manner. Therefore, OptNet is appropriate for embedding the QP-based mapping into the end-to-end training of QPKO. In QPKO, the large-scale QP-based mapping is

decomposed into multiple smaller-scale sub-QP problems, which together form a QP batch. Using OptNet, this batch is solved and back-propagated efficiently on GPU.

## 3. Preliminary

**Koopman Operator Theory-Based Modeling.** Although QPKO is applicable to systems with or without control input, we focus on the controlled case, as it is more general.

A controlled nonlinear dynamic system can be expressed as

$$\mathbf{x}_{k+1} = f(\mathbf{x}_k, \mathbf{u}_k), \tag{1}$$

where $\mathbf{x}_k \in \mathbb{R}^n$ and $\mathbf{u}_k \in \mathbb{R}^m$ are, respectively, the state and input of the system at time step $k$. $f : \mathbb{R}^n \times \mathbb{R}^m \to \mathbb{R}^n$ describes the nonlinear dynamics of the system.

According to the Koopman operator theory (Koopman, 1931) and its extension to controlled systems (Korda & Mezic, 2018; Mauroy et al., 2020), there exists an infinite-dimensional Koopman operator $\mathcal{K} : \mathcal{H} \to \mathcal{H}$ acting on a Hilbert space $\mathcal{H}$ of observable functions, such that the system (1) can be represented in a linear manner, i.e.

$$\mathcal{K}\psi(\begin{bmatrix} \mathbf{x}_k \\ \mathbf{U}_k \end{bmatrix}) = \psi(\begin{bmatrix} f(\mathbf{x}_k, \mathbf{u}_k) \\ \mathcal{S}\mathbf{U}_k \end{bmatrix}) = \psi(\begin{bmatrix} \mathbf{x}_{k+1} \\ \mathbf{U}_{k+1} \end{bmatrix}), \tag{2}$$

where $\psi \in \mathcal{H}$ denotes an observable function, $\mathcal{S}$ denotes the left shift operator ($\mathcal{S}\mathbf{U}_k = \mathbf{U}_{k+1}$), $\mathbf{U}_k = (\mathbf{u}_i)_{i=k}^{\infty}$.

By carrying out a finite-dimensional approximation of the above infinite-dimensional theory for practical use, the widely used formulation of Koopman operator theory-based modeling is given in (Korda & Mezic, 2018), i.e.,

$$\begin{aligned} \mathbf{z}_{k+1} &= \mathbf{A}\mathbf{z}_k + \mathbf{B}\mathbf{u}_k, \\ \mathbf{z}_k &= g(\mathbf{x}_k), \end{aligned} \tag{3}$$

where $g : \mathbb{R}^n \to \mathbb{R}^N$ ($N \gg n$) is formed by stacking the observable functions in this formulation. It maps the original state $\mathbf{x}_k$ to the lifted state $\mathbf{z}_k \in \mathbb{R}^N$. $\mathbf{A} \in \mathbb{R}^{N \times N}$ and $\mathbf{B} \in \mathbb{R}^{N \times m}$ are state transition and input matrices of the global linear model in the lifted space. The objective of the Koopman framework is to identify $\mathbf{A}$, $\mathbf{B}$, and $g$.

**OptNet.** OptNet (Amos & Kolter, 2017) is designed to embed a QP problem into an end-to-end deep learning framework as a differentiable layer, which can be expressed as

$$\begin{aligned} \mathbf{s}^* = \underset{\mathbf{s}}{\operatorname{argmin}} \quad & \frac{1}{2}\mathbf{s}^{\mathrm{T}}\mathbf{Q}(\mathbf{y}_{\mathrm{NN}})\mathbf{s} + \mathbf{q}(\mathbf{y}_{\mathrm{NN}})^{\mathrm{T}}\mathbf{s} \\ \text{s.t.} \quad & \mathbf{V}(\mathbf{y}_{\mathrm{NN}})\mathbf{s} = \mathbf{c}(\mathbf{y}_{\mathrm{NN}}) \\ & \mathbf{G}(\mathbf{y}_{\mathrm{NN}})\mathbf{s} \le \mathbf{h}(\mathbf{y}_{\mathrm{NN}}). \end{aligned} \tag{4}$$

Here, $\mathbf{s}$ is the optimization variable. $\mathbf{Q}$, $\mathbf{q}$, $\mathbf{V}$, $\mathbf{c}$, $\mathbf{G}$, and $\mathbf{h}$ are QP parameters. They are computed from the output

of neural network (NN) $\mathbf{y}_{\mathrm{NN}}$ via differentiable operations. The key feature of OptNet is that it can compute the derivatives of the QP solution $\mathbf{s}^*$ w.r.t. the QP parameters in a numerically stable and computationally efficient manner on GPU. Therefore, this layer can be regarded as a differentiable function of NN output, i.e. $\mathbf{s}^* = f_{\mathrm{QP}}(\mathbf{y}_{\mathrm{NN}})$, which can be embedded in the end-to-end training.

## 4. QPKO: Differentiable QP-Embedded Deep Koopman Framework

The architecture of QPKO is shown in Figure 1. As illustrated in this figure, the core component of QPKO is the QP-based mapping module, which incorporates the mapping from observable functions (parameterized as observable module) to the global linear model in the end-to-end training. In this section, we first detail this module and then describe the end-to-end training of QPKO.

### 4.1. QP-Based Mapping Module

The QP-based mapping module is constructed in two steps. First, from the perspective of both one-step and multi-step prediction accuracy, we formulate a QP-based mapping. Then, an OptNet-based differentiable QP layer is developed to forward and backward propagate this mapping in a numerically stable and computationally efficient manner, yielding the QP-based mapping module.

#### 4.1.1. QP-BASED MAPPING FORMULATION

We first parameterize the observable functions with a trainable observable module, which is the only component with trainable parameters in our QPKO.

**Observable Module.** The input of this module is the original state $\mathbf{x}_k$, while the output is the lifted state $\mathbf{z}_k$. As shown in Figure 1, this module consists of two parts, i.e., an identity mapping and an MLP. The MLP is adopted to generate the majority of the lifted state, while identity mapping is to incorporate the original state into the lifted state. This incorporation allows the control scheme to be directly designed in the lifted space. Overall, this module can be expressed as

$$\mathbf{z}_k = g(\mathbf{x}_k; \boldsymbol{\theta}) = [\mathbf{x}_k, g_{\mathrm{MLP}}(\mathbf{x}_k; \boldsymbol{\theta})]^{\mathrm{T}}, \tag{5}$$

where $g(\,\cdot\,; \boldsymbol{\theta})$ denotes the observable module, $g_{\mathrm{MLP}}$ denotes the forward propagation of the MLP, $\boldsymbol{\theta}$ collects the MLP's trainable parameters.

**QP-Based Mapping.** Then, we formulate the mapping from observable functions to the global linear model as a QP problem, consisting of a one-step accuracy-oriented objective function and a set of multi-step accuracy-oriented constraints. This formulation requires a QP dataset $\mathcal{D}_{\mathrm{QP}}$, which

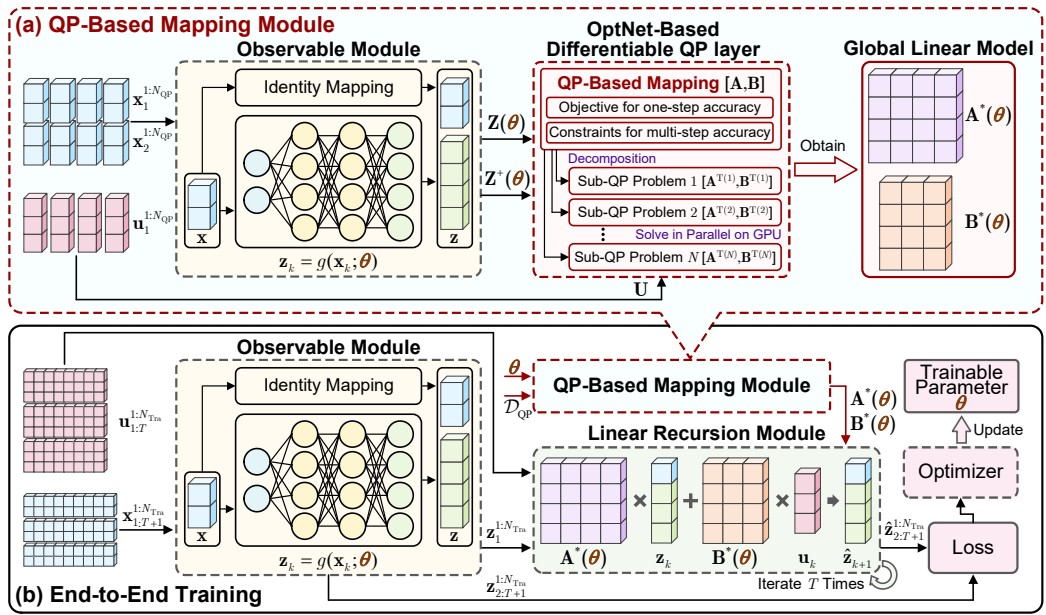

*Figure 1.* The architecture of QPKO. The core component of QPKO is the QP-based mapping module, which is detailed in part (a). Through this module, the mapping from observable functions (parameterized by the observable module with trainable parameters $\boldsymbol{\theta}$) to the global linear model is incorporated into the end-to-end training of QPKO, as shown in part (b). Specifically, at each training iteration, this module computes the global linear model (i.e., $\mathbf{A}^*(\boldsymbol{\theta})$ and $\mathbf{B}^*(\boldsymbol{\theta})$) based on the current learned observable module. The resulting model is then fed into the linear recursion module to carry out $T$-step prediction for loss evaluation and end-to-end optimization of $\boldsymbol{\theta}$.

comprises $N_{\text{QP}}$ one-step transition tuples of the modeled nonlinear system, i.e., $\mathcal{D}_{\text{QP}} = \{\mathbf{x}_1^{1:N_{\text{QP}}}, \mathbf{x}_2^{1:N_{\text{QP}}}, \mathbf{u}_1^{1:N_{\text{QP}}}\}$.

*One-Step Accuracy-Oriented Objective Function*: Based on $\mathcal{D}_{\text{QP}}$ and the observable module, the objective function of this QP problem is formulated as

$$\min_{\mathbf{A},\mathbf{B}} \left\| \mathbf{Z}\left(\mathcal{D}_{\text{QP}};\boldsymbol{\theta}\right)\mathbf{A}^{\text{T}} + \mathbf{U}(\mathcal{D}_{\text{QP}})\mathbf{B}^{\text{T}} - \mathbf{Z}^+\left(\mathcal{D}_{\text{QP}};\boldsymbol{\theta}\right) \right\|_{\text{F}}^2,$$
(6)

where

$$\mathbf{Z}\left(\mathcal{D}_{\text{QP}};\boldsymbol{\theta}\right) := \left[g(\mathbf{x}_1^1;\boldsymbol{\theta}), \cdots, g(\mathbf{x}_1^{N_{\text{QP}}};\boldsymbol{\theta})\right]^{\text{T}},$$

$$\mathbf{Z}^+\left(\mathcal{D}_{\text{QP}};\boldsymbol{\theta}\right) := \left[g(\mathbf{x}_2^1;\boldsymbol{\theta}), \cdots, g(\mathbf{x}_2^{N_{\text{QP}}};\boldsymbol{\theta})\right]^{\text{T}}, \quad (7)$$

$$\mathbf{U}(\mathcal{D}_{\text{QP}}) := \left[\mathbf{u}_1^1, \cdots, \mathbf{u}_1^{N_{\text{QP}}}\right]^{\text{T}}.$$

Since $\mathcal{D}_{\text{QP}}$ is typically kept fixed during training and QPKO optimizes only $\boldsymbol{\theta}$, we use the following shorthand for notational simplicity: $\mathbf{Z}(\mathcal{D}_{\text{QP}};\boldsymbol{\theta}) := \mathbf{Z}(\boldsymbol{\theta})$, $\mathbf{Z}^+(\mathcal{D}_{\text{QP}};\boldsymbol{\theta}) := \mathbf{Z}^+(\boldsymbol{\theta})$, and $\mathbf{U}(\mathcal{D}_{\text{QP}}) := \mathbf{U}$.

*Multi-Step Accuracy-Oriented Constraints*: The objective function (6) only minimizes the one-step prediction error, while the overall objective of QPKO is to reduce the multi-step prediction error. To mitigate this objective mismatch, we design a set of multi-step accuracy-oriented constraints for the QP-based mapping. This design is motivated by the following theorem (whose proof is given in Appendix B):

**Theorem 4.1.** *For the global linear model parameterized by $\mathbf{A}$ and $\mathbf{B}$, if $\|\mathbf{A}\|_\infty < 1$, then its cumulative prediction error over any prediction horizon is bounded. Here, $\|\cdot\|_\infty$ denotes the infinity norm.*

According to this theorem, we can enhance the multi-step prediction accuracy by letting $\|\mathbf{A}\|_\infty < 1$. However, this constraint is nonlinear, which cannot be incorporated into the QP problem directly. To address this, we transform it into an equivalent set of linear constraints, that is

$$\sum_{j=1}^{N} r_{i,j} \le 1 - c, \ \ i = 1, \ldots, N,$$
$$-r_{i,j} \le a_{i,j} \le r_{i,j}, \ \ i = 1, \ldots, N, \ j = 1, \ldots, N,$$
(8)

where $a_{i,j}$ denotes the $(i,j)$-entry of $\mathbf{A}$, $r_{i,j}$ denotes the auxiliary decision variable introduced in this transformation, $c$ is a very small positive constant used to replace $<$ with $\le$ for QP formulation.

It is noteworthy that the boundedness result in Theorem 4.1 can be established under several types of matrix norms. In this work, the infinity norm is adopted for two reasons: (i) the resulting norm-based constraint can be equivalently reformulated into linear constraints, as shown in (8). This preserves the QP formulation. (ii) Under these linear constraints, the QP-based mapping can be equivalently decomposed into multiple independent sub-QP problems, which will be detailed in Section 4.1.2. This decomposition improves the scalability of QPKO.

*QP-Based Mapping*: The objective function (6) and constraints (8) together constitute the QP-based mapping, i.e.

$$(\mathbf{A}^*, \mathbf{B}^*, r_{i,j}^*, \xi_i^*) =$$

$$\underset{\mathbf{A}, \mathbf{B}, r_{i,j}, \xi_i}{\operatorname{argmin}} \left\| \mathbf{Z}(\boldsymbol{\theta}) \mathbf{A}^{\mathrm{T}} + \mathbf{U} \mathbf{B}^{\mathrm{T}} - \mathbf{Z}^+(\boldsymbol{\theta}) \right\|_{\mathrm{F}}^2 + \alpha \sum_{i=1}^N \xi_i^2$$

$$\text{s.t. } \sum_{j=1}^N r_{i,j} + \xi_i \le 1 - c, i = 1, \dots, N,$$

$$- r_{i,j} \le a_{i,j} \le r_{i,j}, \ i = 1, \dots, N, j = 1, \dots, N. \tag{9}$$

Given the current learned observable functions (parameterized by $\boldsymbol{\theta}$), this mapping computes the corresponding global linear model (i.e., $\mathbf{A}^*$ and $\mathbf{B}^*$) by jointly considering one-step and multi-step prediction accuracy. Moreover, one may notice that a set of slack variables, i.e., $\{\xi_i\}_{i=1}^N$, is additionally introduced in this QP problem to relax the constraints (8). The rationale behind this design is as follows: the constraints (8) are not designed to directly minimize the multi-step prediction error. Instead, they work as a regularizer that biases the QP-based mapping toward generating a stable global linear model, which is less likely to amplify rollout errors over long prediction horizons. However, for a class of nonlinear systems whose lifted dynamics are inherently marginally stable or unstable, this regularizer may limit the expressiveness of the resulting global linear model. To address this issue, we introduce the slack variables to relax the constraints (8) and control this relaxation using a penalty weight, i.e., $\alpha$. This weight is treated as a tunable hyperparameter.

### 4.1.2. QP-BASED MAPPING MODULE DEVELOPMENT

Equation (9) specifies the forward-propagation of the QP-based mapping module. It is carried out in two steps: (i) for given $\boldsymbol{\theta}$ and $\mathcal{D}_{\mathrm{QP}}$, calculate $\mathbf{Z}$ and $\mathbf{Z}^+$ via the observable module, while constructing $\mathbf{U}$ via a stacking operation, as detailed in (7); (ii) based on $\mathbf{Z}$, $\mathbf{Z}^+$, and $\mathbf{U}$, solve the QP problem in (9), yielding $\mathbf{A}^*$ and $\mathbf{B}^*$.

However, to enable end-to-end training of QPKO, this module must support numerically stable back-propagation. This requires that Jacobians $\frac{\partial \mathrm{vec}(\mathbf{A}^*)}{\partial \boldsymbol{\theta}}$ and $\frac{\partial \mathrm{vec}(\mathbf{B}^*)}{\partial \boldsymbol{\theta}}$ can be computed in a numerically stable manner. Based on the chain rule, these Jacobians can be detailed as

$$\frac{\partial \mathrm{vec}(\mathbf{A}^*)}{\partial \boldsymbol{\theta}} = \frac{\partial \mathrm{vec}(\mathbf{A}^*)}{\partial \mathrm{vec}(\mathbf{Z})} \frac{\partial \mathrm{vec}(\mathbf{Z})}{\partial \boldsymbol{\theta}} + \frac{\partial \mathrm{vec}(\mathbf{A}^*)}{\partial \mathrm{vec}(\mathbf{Z}^+)} \frac{\partial \mathrm{vec}(\mathbf{Z}^+)}{\partial \boldsymbol{\theta}},$$

$$\frac{\partial \mathrm{vec}(\mathbf{B}^*)}{\partial \boldsymbol{\theta}} = \frac{\partial \mathrm{vec}(\mathbf{B}^*)}{\partial \mathrm{vec}(\mathbf{Z})} \frac{\partial \mathrm{vec}(\mathbf{Z})}{\partial \boldsymbol{\theta}} + \frac{\partial \mathrm{vec}(\mathbf{B}^*)}{\partial \mathrm{vec}(\mathbf{Z}^+)} \frac{\partial \mathrm{vec}(\mathbf{Z}^+)}{\partial \boldsymbol{\theta}}. \tag{10}$$

In this equation, the Jacobians $\frac{\partial \mathrm{vec}(\mathbf{Z})}{\partial \boldsymbol{\theta}}$ and $\frac{\partial \mathrm{vec}(\mathbf{Z}^+)}{\partial \boldsymbol{\theta}}$ are the back-propagation results of the step (i). They can be readily computed via automatic differentiation, since the step (i) only involves the forward-propagation of the MLP-based observable module and simple stacking operations.

In contrast, computing the Jacobians $\frac{\partial \mathrm{vec}(\mathbf{A}^*)}{\partial \mathrm{vec}(\mathbf{Z})}$, $\frac{\partial \mathrm{vec}(\mathbf{A}^*)}{\partial \mathrm{vec}(\mathbf{Z}^+)}$, $\frac{\partial \mathrm{vec}(\mathbf{B}^*)}{\partial \mathrm{vec}(\mathbf{Z})}$, and $\frac{\partial \mathrm{vec}(\mathbf{B}^*)}{\partial \mathrm{vec}(\mathbf{Z}^+)}$, i.e., back-propagating through the step (ii), is much more challenging, since it essentially amounts to differentiating the QP solution (i.e., $\mathbf{A}^*$ and $\mathbf{B}^*$) w.r.t. the QP parameters (i.e., $\mathbf{Z}$ and $\mathbf{Z}^+$).

As discussed in Sections 2 and 3, OptNet is appropriate to address this issue, as it can back-propagate through QPs in a numerically stable manner. Therefore, based on OptNet, we build a differentiable QP layer to implement step (ii), i.e., solving and back-propagating the QP problem in (9).

**OptNet-Based differentiable QP Layer.** The QP problem in (9) is large-scale, with $2N^2 + Nm + N$ decision variables and $2N^2 + N$ constraints. Solving and back-propagating it directly will be computationally expensive and degrade the scalability of QPKO. OptNet's capability in GPU-based batched QP solving and back-propagation opens the possibility to address this issue. To leverage this capability, our differentiable QP layer does not solve and back-propagate this problem as a whole. Instead, we equivalently decompose it into $N$ independent sub-QP problems with $2N + m + 1$ decision variables and $2N + 1$ constraints. These sub-QP problems together constitute a QP batch, which is solved and back-propagated by OptNet in parallel on GPU. This design can effectively increase the training efficiency.

The $h$-th sub-QP problem can be expressed as

$$\min_{\mathbf{A}^{\mathrm{T}(h)}, \mathbf{B}^{\mathrm{T}(h)}, r_{h,j}, \xi_h} \left\| \mathbf{Z}(\boldsymbol{\theta}) \mathbf{A}^{\mathrm{T}(h)} + \mathbf{U} \mathbf{B}^{\mathrm{T}(h)} - \mathbf{Z}^{+(h)}(\boldsymbol{\theta}) \right\|_2^2$$
$$+ \alpha \xi_h^2$$

$$\text{s.t.} \quad \sum_{j=1}^N r_{h,j} + \xi_h \le 1 - c,$$

$$- r_{h,j} \le a_{h,j} \le r_{h,j}, \ j = 1, \dots, N. \tag{11}$$

Here, $\mathbf{A}^{\mathrm{T}(h)} \in \mathbb{R}^N$ and $\mathbf{B}^{\mathrm{T}(h)} \in \mathbb{R}^m$, and $\mathbf{Z}^{+(h)} \in \mathbb{R}^{N_{\mathrm{QP}}}$, respectively, denote the $h$-th column of $\mathbf{A}^{\mathrm{T}}$, $\mathbf{B}^{\mathrm{T}}$, and $\mathbf{Z}^+$.

**Proposition 4.2.** *The QP problem in (9) is equivalent to the $N$ independent sub-QP problems in (11). The proof is given in Appendix C.*

For brevity, we denote the OptNet-based differentiable QP layer $f_{\mathrm{QP}}$ as

$$(\mathbf{A}^*(\boldsymbol{\theta}), \mathbf{B}^*(\boldsymbol{\theta})) = f_{\mathrm{QP}}(\mathbf{Z}(\boldsymbol{\theta}), \mathbf{Z}^+(\boldsymbol{\theta})), \tag{12}$$

$\mathbf{A}^*(\boldsymbol{\theta})$ and $\mathbf{B}^*(\boldsymbol{\theta})$ are shorthand for $\mathbf{A}^*(\mathcal{D}_{\mathrm{QP}}; \boldsymbol{\theta})$ and $\mathbf{B}^*(\mathcal{D}_{\mathrm{QP}}; \boldsymbol{\theta})$. The leveraging of OptNet enables the forward and backward propagation of this layer to be conducted in a numerically stable and computationally efficient manner.

**QP-Based Mapping Module.** Eventually, as shown in Figure 1, the observable module and the OptNet-based differentiable QP layer together constitute the QP-based mapping

*Table 1.* The NMAE given by each framework across four nonlinear dynamical systems and prediction horizons $T \in \{50, 100, 150, 200\}$. Results are reported as mean$\pm$std (in units of $10^{-2}$). The mean and standard deviation are computed over 5 independent training trials. The best-performing results are highlighted in **bold**, while the second-best results are underlined.

| | CARTPOLE ($\times 10^{-2}$) | | | | VDP ($\times 10^{-2}$) | | | |
|---|---|---|---|---|---|---|---|---|
| FRAMEWORK | $T = 50$ | $T = 100$ | $T = 150$ | $T = 200$ | $T = 50$ | $T = 100$ | $T = 150$ | $T = 200$ |
| DKO | $0.541\pm0.071$ | $1.080\pm0.134$ | $2.092\pm0.246$ | $3.361\pm0.340$ | $0.206\pm0.036$ | $0.491\pm0.042$ | $1.480\pm0.334$ | $2.492\pm0.693$ |
| EDMD-DL | $0.397\pm0.097$ | $1.004\pm0.188$ | $1.975\pm0.256$ | $3.133\pm0.351$ | $0.180\pm0.032$ | $0.456\pm0.041$ | $\underline{1.145}\pm0.246$ | $\underline{1.891}\pm0.448$ |
| NDMD | $\underline{0.360}\pm0.056$ | $\underline{0.766}\pm0.143$ | $\underline{1.460}\pm0.287$ | $\underline{2.473}\pm0.443$ | $\underline{0.118}\pm0.026$ | $\underline{0.389}\pm0.079$ | $1.317\pm0.282$ | $2.259\pm0.503$ |
| QPKO | $\mathbf{0.325}\pm\mathbf{0.057}$ | $\mathbf{0.663}\pm\mathbf{0.096}$ | $\mathbf{1.073}\pm\mathbf{0.165}$ | $\mathbf{1.783}\pm\mathbf{0.257}$ | $\mathbf{0.111}\pm\mathbf{0.019}$ | $\mathbf{0.330}\pm\mathbf{0.082}$ | $\mathbf{0.885}\pm\mathbf{0.173}$ | $\mathbf{1.640}\pm\mathbf{0.375}$ |

| | WT ($\times 10^{-2}$) | | | | TSR ($\times 10^{-2}$) | | | |
|---|---|---|---|---|---|---|---|---|
| FRAMEWORK | $T = 50$ | $T = 100$ | $T = 150$ | $T = 200$ | $T = 50$ | $T = 100$ | $T = 150$ | $T = 200$ |
| DKO | $1.215\pm0.194$ | $1.350\pm0.170$ | $1.717\pm0.229$ | $2.146\pm0.331$ | $0.583\pm0.130$ | $0.765\pm0.119$ | $1.309\pm0.260$ | $\underline{1.946}\pm0.345$ |
| EDMD-DL | $\underline{0.509}\pm0.097$ | $\underline{0.876}\pm0.201$ | $\underline{1.275}\pm0.319$ | $\underline{1.675}\pm0.447$ | $0.630\pm0.112$ | $0.948\pm0.246$ | $1.603\pm0.518$ | $2.355\pm1.052$ |
| NDMD | $0.513\pm0.155$ | $0.928\pm0.336$ | $1.381\pm0.503$ | $1.853\pm0.636$ | $\underline{0.275}\pm0.130$ | $\underline{0.445}\pm0.142$ | $\underline{1.222}\pm0.353$ | $2.315\pm0.763$ |
| QPKO | $\mathbf{0.506}\pm\mathbf{0.090}$ | $\mathbf{0.841}\pm\mathbf{0.188}$ | $\mathbf{1.174}\pm\mathbf{0.277}$ | $\mathbf{1.520}\pm\mathbf{0.347}$ | $\mathbf{0.142}\pm\mathbf{0.056}$ | $\mathbf{0.255}\pm\mathbf{0.068}$ | $\mathbf{0.649}\pm\mathbf{0.165}$ | $\mathbf{1.147}\pm\mathbf{0.347}$ |

module $f_{\mathrm{map}}$. For brevity, we denote this module as

$$(\mathbf{A}^*(\boldsymbol{\theta}), \mathbf{B}^*(\boldsymbol{\theta})) = f_{\mathrm{map}}(\boldsymbol{\theta}), \tag{13}$$

Notably, $f_{\mathrm{map}}(\boldsymbol{\theta})$ is shorthand for $f_{\mathrm{map}}(\mathcal{D}_{\mathrm{QP}}; \boldsymbol{\theta})$. This module can be embedded into the end-to-end training of QPKO, as it can back-propagate (i.e., compute the Jacobians $\frac{\partial \mathrm{vec}(\mathbf{A}^*)}{\partial \boldsymbol{\theta}}$ and $\frac{\partial \mathrm{vec}(\mathbf{B}^*)}{\partial \boldsymbol{\theta}}$) in a numerically stable manner.

### 4.2. End-to-End Training of QPKO

As shown in Figure 1, besides the observable module and QP-based mapping module introduced in Section 4.1, the linear recursion module also participates in QPKO's end-to-end training. Next, we detail this module.

**Linear Recursion Module.** After receiving the global linear model, i.e., $\mathbf{A}^*(\boldsymbol{\theta})$ and $\mathbf{B}^*(\boldsymbol{\theta})$ from the QP-based mapping module, the linear recursion module conducts $T$-step prediction, i.e.

$$\begin{cases} \hat{\mathbf{z}}_1 = \mathbf{z}_1, \\ \hat{\mathbf{z}}_{k+1} = \mathbf{A}^*(\boldsymbol{\theta})\hat{\mathbf{z}}_k + \mathbf{B}^*(\boldsymbol{\theta})\mathbf{u}_k, \ \ k = 1, \ldots, T, \end{cases} \tag{14}$$

where $\hat{\mathbf{z}}_k$ denotes the predicted value of the lifted state at time step $k$. The true value $\mathbf{z}_1$ is computed by the observable module (5).

**Loss Function.** The loss function $\mathcal{L}$ of this training is designed to quantify $T$-step rollout error, i.e.

$$\mathcal{L} = \sum_{k=1}^{T} \|\hat{\mathbf{z}}_{k+1} - \mathbf{z}_{k+1}\|_2^2. \tag{15}$$

**Training Procedure.** The training of QPKO is essentially to optimize the trainable parameters $\boldsymbol{\theta}$ by minimizing the loss function (15), i.e., $\min_{\boldsymbol{\theta}} \mathcal{L}(\boldsymbol{\theta})$. The pseudocode of QPKO's end-to-end training procedure is given in Appendix D.

## 5. Experiments

In this section, we evaluate QPKO in terms of **modeling accuracy**, **training efficiency**, and **control performance** on four controlled nonlinear dynamical systems, i.e., the forced Van der Pol oscillator (VDP), CartPole, wind turbine (WT), and tethered space robot (TSR). Moreover, an ablation study is conducted to evaluate the effect of the multi-step accuracy-oriented constraints (8) on QPKO's modeling accuracy. The results are reported in Appendix F.

For each system, we collect an operational dataset consisting of 100 trajectories with a length of 200 time steps. For CartPole and VDP, both the initial state and the control input are randomly generated. For WT and TSR, their initial state is randomly sampled, while their control input is generated differently. Specifically, for WT, it is obtained based on the operational data collected at a real-world wind farm. For TSR, it is produced by a predetermined control law. In each dataset, 70, 15, and 15 trajectories are, respectively, adopted for training, validation, and testing. The four systems and dataset collection are detailed in Appendix E.1.

We compare QPKO with three representative deep learning-based Koopman frameworks spanning the two training paradigms discussed in Section 1, i.e., DKO (Shi & Meng, 2022), EDMD-DL (Han et al., 2020), and NDMD (Iwata & Kawahara, 2023). Among them, DKO adopts the first training paradigm, where the global linear model is treated as an independent trainable component. EDMD-DL and NDMD both follow the second training paradigm, where the model is computed by an LS problem. More specifically, EDMD-DL is trained in two alternating stages, whereas the training of NDMD is conducted in an end-to-end manner. Notably, although the above frameworks exhibit distinct training paradigms, the global linear models established by them share the same structure as defined in (3).

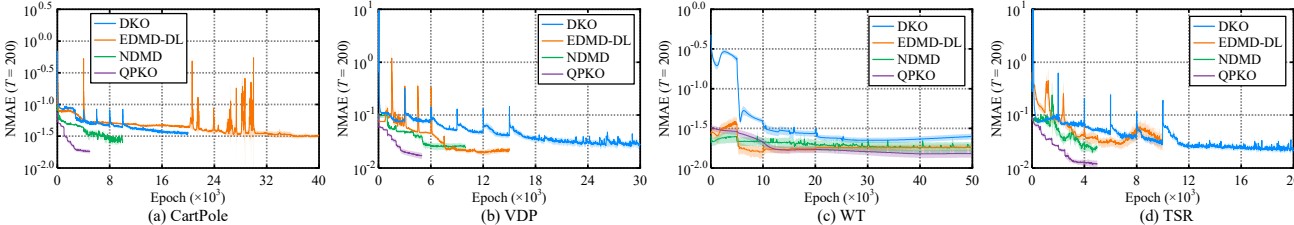

*Figure 2.* Per-step prediction error over a prediction horizon $T = 200$. The solid lines and shaded regions, respectively, represent the mean and $\pm 0.5 \times$ standard deviation over 5 independent training trials.

*Figure 3.* NMAE for prediction horizon $T = 200$ as a function of training epoch. The solid lines and shaded regions, respectively, represent the mean and $\pm 0.5 \times$ standard deviation over 5 independent training trials.

Moreover, all frameworks follow a curriculum learning strategy (Sakib & Pan, 2025): we initialize the $T$ in the loss function (15) to 10 and increase it by 15 every $0.1 Epoch_{max}$, until it reaches the maximum value of 85 at $0.5 Epoch_{max}$. For a fair training efficiency comparison, we select the hyperparameters of the three baselines on the principle of accelerating training while preserving validation performance. Hyperparameters are listed in Appendix E.2.

### 5.1. Modeling Accuracy Evaluation

We quantify the modeling accuracy using the normalized mean absolute error (NMAE), which is defined in Appendix E.3. Table 1 reports the NMAE for multi-step predictions with horizons $T \in \{50, 100, 150, 200\}$.

This table shows that QPKO achieves the lowest NMAE across all systems and horizons, indicating its superiority in modeling accuracy. This superiority is particularly pronounced for long-horizon prediction, i.e., $T \in \{150, 200\}$. Specifically, on TSR, QPKO reduces the mean NMAE by 46.9% and 41.1% at $T = 150$ and $T = 200$ compared to the second-best framework. On CartPole, VDP, and WT, the corresponding reductions are (26.5%, 27.9%), (22.7%, 13.3%), and (7.9%, 9.3%), respectively. One potential reason for this superiority is that QPKO computes the global linear model via a QP-based mapping module, which considers both one-step and multi-step prediction accuracy. This convex optimization-based global linear model computation can lead to a reliable modeling accuracy.

Figure 2 plots the per-step prediction error (defined in Appendix E.4) over a prediction horizon of $T = 200$. As shown in this figure, due to error accumulation, the per-step

prediction error of all frameworks shows an increasing trend with the prediction step. Nevertheless, QPKO exhibits the slowest error growth, and its error curves are almost always the lowest across all systems and prediction steps. This result illustratively shows the improvement of modeling accuracy, especially the long-term one, given by QPKO.

Moreover, further evaluations regarding (i) the modeling accuracy in the presence of measurement noise and (ii) the sensitivity of QPKO to the slack penalty weight $\alpha$ used in the QP-based mapping (9) are detailed in Appendices G.1 and G.2, respectively.

### 5.2. Training Efficiency Evaluation

Table 2 reports the wall-clock training time of each framework. The training time is defined as the time to reach the best validation performance in a training trial. According to this table, on CartPole, VDP, and TSR, QPKO requires the shortest training time, yielding a 3.1×, 2.6×, and 1.9× speedup over the second-best framework, respectively. On WT, EDMD-DL is faster than QPKO. Nevertheless, all methods incur relatively short training times on this system. As real-world wind farms often operate under maximum power point tracking mode, this phenomenon may be due to the narrow range of operating conditions covered by the collected WT data, which reduces the training difficulty. Therefore, from the overall perspective, QPKO requires shorter training time. Combining this result with QPKO's higher modeling accuracy discussed in Section 5.1, we draw the conclusion that QPKO exhibits higher training efficiency.

Moreover, combining the results on the four systems, DKO and EDMD-DL, respectively, exhibit the longest and the

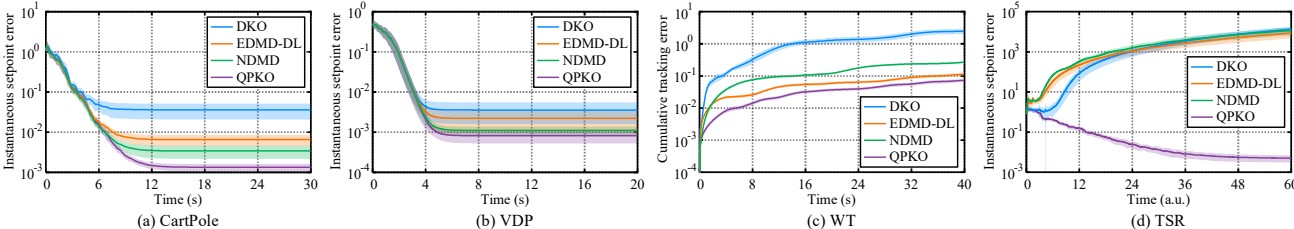

*Figure 4.* Instantaneous setpoint error / cumulative tracking error as a function of time. Solid lines and shaded regions, respectively, denote the mean and $\pm 0.5 \times$ standard deviation over 50 control trials (10 initial states / tracking references $\times$ 5 independent training trials).

most variable training time. This is because (i) DKO treats the observable functions and the global linear model as two independent trainable components, which increases the optimization complexity. (ii) EDMD-DL is trained in an alternating two-stage manner, which is prone to unstable convergence. Conversely, QPKO introduces a differentiable QP-based mapping module to compute the global linear model from the learned observable functions and enable end-to-end training, yielding its superior training efficiency.

*Table 2.* Wall-clock training time of each framework. Results are reported as mean±std (in hours) over 5 independent training trials. The best results are highlighted in **bold**, while the second-best results are underlined.

| FRAMEWORK | CARTPOLE | VDP | WT | TSR |
|---|---|---|---|---|
| DKO | 3.80±0.25 | 5.43±0.13 | 0.74±0.22 | 4.08±0.23 |
| EDMD-DL | 7.05±0.64 | 1.86±0.23 | **0.23±0.27** | 1.37±0.42 |
| NDMD | 1.85±0.13 | 1.40±0.39 | 0.70±0.24 | 1.04±0.02 |
| QPKO | **0.60±0.01** | **0.55±0.03** | 0.56±0.02 | **0.56±0.04** |

To further illustrate the training efficiency, we compute the NMAE for $T = 200$ on the test set and plot it as a function of the training epoch in Figure 3. Notably, these NMAE curves are only adopted to demonstrate the training efficiency and not used for model selection or hyperparameter tuning. Moreover, the regularly spaced NMAE spikes and abrupt changes during the first half of training are caused by the curriculum learning strategy, i.e., the periodic increase of $T$ within the loss function (15).

As shown in Figure 3, on CartPole, VDP, and TSR, the NMAE curve given by QPKO is always the lowest and decreases much faster with epoch. On WT, EDMD-DL decreases the error faster in the early epochs, whereas QPKO reaches a lower final NMAE. Besides, QPKO's NMAE converges most stably across the four systems. A salient contrast is observed on CartPole, where EDMD-DL's alternating two-stage training leads to noticeable NMAE spikes in the latter half of training. These results illustratively demonstrate the superiority of QPKO in training efficiency. It is noteworthy that this demonstration is conservative, as QPKO is trained in a full-batch setting, while the baselines are trained with mini-batches. That is, QPKO performs far

fewer training iterations per epoch than the baselines.

Additional training efficiency results, including (i) the per-epoch training time of each framework and (ii) the computational cost of QPKO under different lifted dimensions, are reported in Appendices H.1 and H.2, respectively.

### 5.3. Control Performance Evaluation

Finally, we evaluate the global linear models established by each framework in model predictive control (MPC). The rationale for this evaluation is threefold. (i) MPC is one of the primary downstream applications of Koopman operator theory-based modeling. (ii) MPC relies on accurate multi-step predictions to make control decisions, and insufficient prediction accuracy will lead to control failure. Therefore, this evaluation provides a stricter test of model accuracy, especially the multi-step accuracy. (iii) The data distribution induced by MPC can differ from that of our offline dataset, so this evaluation also assesses the generalization performance of the models.

*Table 3.* The final values of setpoint error or cumulative tracking error. Results are reported as mean±std over 50 control trials. The best results are highlighted in **bold**, while the second-best results are underlined.

| | FINAL SETPOINT ERROR $(\times 10^{-2})$ | | | FINAL CUMULATIVE TRACKING ERROR |
|---|---|---|---|---|
| FRAMEWORK | CARTPOLE | VDP | TSR | WT |
| DKO | 3.61±3.06 | 0.36±0.40 | – | 2.47±0.90 |
| EDMD-DL | 0.65±0.40 | 0.22±0.25 | – | 0.11±0.02 |
| NDMD | 0.34±0.25 | 0.11±0.06 | – | 0.27±0.04 |
| QPKO | **0.13±0.06** | **0.08±0.06** | **0.51±0.39** | **0.07±0.02** |

Due to the control failure, the results given by DKO, EDMD-DL, and NDMD on TSR are denoted as –.

As for the control objective, on CartPole, VDP, and TSR, we implement MPC to drive the system from a randomly sampled initial state to the origin. On WT, MPC coordinates the active power and pitch angle under the time-varying wind speed to make WT's rotor speed track a sinusoidal reference with a period of 20 s and randomly sampled parameters. For each system, we sample 10 initial states (or 10 tracking references). This sampling and additional details of the MPC implement are provided in Appendix E.5.

To quantify control performance, on CartPole, VDP, and TSR we compute the instantaneous setpoint error, while on WT we report the cumulative tracking error. These metrics are detailed in Appendix E.6. Figure 4 plots these metrics as a function of time, and Table 3 reports their final values.

According to Figure 4, QPKO clearly outperforms the baselines in control performance. This advantage is most pronounced on TSR, where MPC controllers based on the three baselines fail to drive the state to the origin. Such control failures are typically caused by model mismatch, arising from insufficient model accuracy and limited generalization under the data distribution shift induced by closed-loop control. Conversely, the MPC built on QPKO successfully achieves the control objective. Moreover, on CartPole and VDP, the setpoint error given by QPKO converges to a smaller steady-state value, while on WT, QPKO consistently attains the lowest cumulative tracking error. These results indicate that QPKO can achieve more accurate setpoint regulation and reference tracking. Finally, Table 3 shows that, on CartPole, VDP, and TSR, QPKO reduces the final control error by 61.8%, 27.3%, and 36.4%, respectively, compared with the second-best baseline. QPKO's superior MPC performance can be attributed to the improved accuracy and generalization capability of the learned model.

## 6. Conclusion

In this work, we propose a differentiable QP-embedded deep Koopman framework (QPKO), whose key features are twofold. (i) Using a QP-based mapping module, it computes the global linear model based on the current learned observable functions from a perspective of both single-step and multi-step accuracy. This can reduce the training complexity and lead to reliable modeling accuracy. (ii) It supports end-to-end training, which facilitates stable convergence. Experiments on four nonlinear dynamical systems illustrate QPKO's superiority in modeling accuracy, training efficiency, and control performance.

Nevertheless, we would like to highlight two limitations here. (i) As detailed in Section 4.1.1, constraints (8) do not directly minimize the multi-step prediction error. Instead, they enhance the multi-step accuracy in an indirect manner by biasing toward a stable global linear model. A mechanism that enables the QP-based mapping to directly reduce the multi-step prediction error is desired. (ii) The current experiments are conducted on ODE-based systems whose dimensions are relatively low. The scalability of QPKO to high-dimensional systems, such as PDE-based systems, should be further investigated.

In addition to addressing the above limitations, future work will also focus on incorporating physical information into QPKO to reduce its data dependency.

## Acknowledgements

The authors would like to thank the anonymous reviewers for their constructive feedback. This work was supported by the National Natural Science Foundation of China (Grant No. 52477062).

## Impact Statement

The goal of this work is to advance the field of deep learning-based Koopman modeling, which can benefit the modeling, control, and analysis of nonlinear dynamical systems. There are many potential societal consequences of our work, none of which we feel must be specifically highlighted here.

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

## A. Additional Related Work

**Koopman Operator theory and Differentiable Convex Optimization-Based Policy Learning.** Similar to QPKO, several works in the policy learning field also leverage both Koopman operator theory and differentiable convex optimization (DCO). To avoid potential confusion, we briefly clarify them here. In these works, the policy network is formulated in the structure of a Koopman model-based LQR (Yin et al., 2022) or MPC (Mayfrank et al., 2025; Retchin et al., 2023) controller, and DCO is leveraged to make this optimal controller-structured policy network differentiable. Then, this network can be trained via imitation learning (Yin et al., 2022) or deep reinforcement learning (Yin et al., 2022; Mayfrank et al., 2025; Retchin et al., 2023). Overall, these works aim to learn a policy that maps observations to control inputs, and DCO here is adopted to differentiate through this policy. In contrast, our QPKO targets system modeling, i.e., learning a global linear model for nonlinear dynamical systems. Here, the DCO method (specifically, OptNet (Amos & Kolter, 2017)) is leveraged to forward and backward propagate the developed QP-based mapping in a numerically stable and computationally efficient manner.

## B. Proof of Theorem 4.1

**Theorem B.1** (Restatement of Theorem 4.1). *For the global linear model parameterized by $\mathbf{A}$ and $\mathbf{B}$, if $\|\mathbf{A}\|_\infty < 1$, then its cumulative prediction error over any prediction horizon is bounded. Here, $\|\cdot\|_\infty$ denotes the infinity norm.*

*Proof.* We denote $\varepsilon_k$ as the one-step prediction error of this global linear model at time step $k$. Then, the first-step prediction conducted by this model can be expressed as

$$\mathbf{z}_{k+1} + \varepsilon_{k+1} = \mathbf{A}\mathbf{z}_k + \mathbf{B}\mathbf{u}_k \tag{16}$$

On the basis of the first-step prediction, the second-step prediction can be expressed as

$$\mathbf{z}_{k+2} + \varepsilon_{k+2} + \mathbf{A}\varepsilon_{k+1} = \mathbf{A}(\mathbf{z}_{k+1} + \varepsilon_{k+1}) + \mathbf{B}\mathbf{u}_{k+1} \tag{17}$$

Iterating the above process, the formulation of the $T$-th-step prediction is derived, i.e.

$$\mathbf{z}_{k+T} + \sum_{i=0}^{T-1} \mathbf{A}^i \varepsilon_{k+T-i} = \mathbf{A}\left(\mathbf{z}_{k+T-1} + \sum_{i=0}^{T-2} \mathbf{A}^i \varepsilon_{k+T-i-1}\right) + \mathbf{B}\mathbf{u}_{k+T-1}. \tag{18}$$

From (18), the cumulative error at the $T$-th step prediction $\mathbf{e}_T$ is given by

$$\mathbf{e}_T = \sum_{i=0}^{T-1} \mathbf{A}^i \varepsilon_{k+T-i}. \tag{19}$$

Then, we adopt the infinity norm to quantify $\mathbf{e}_T$, that is

$$\|\mathbf{e}_T\|_\infty = \left\|\sum_{i=0}^{T-1} \mathbf{A}^i \varepsilon_{k+T-i}\right\|_\infty. \tag{20}$$

Using the triangle inequality and submultiplicativity, the following inequality holds:

$$\|\mathbf{e}_T\|_\infty = \left\|\sum_{i=0}^{T-1} \mathbf{A}^i \varepsilon_{k+T-i}\right\|_\infty \leq \sum_{i=0}^{T-1} \left\|\mathbf{A}^i \varepsilon_{k+T-i}\right\|_\infty \leq \sum_{i=0}^{T-1} \|\mathbf{A}\|_\infty^i \|\varepsilon_{k+T-i}\|_\infty. \tag{21}$$

Then, we denote the maximum magnitude of the one-step prediction error as $\varepsilon_{\max}$. That is, $\varepsilon_{\max} \geq \|\varepsilon_{k+T-i}\|_\infty$ for $i = 0, 1, \ldots, T-1$. Subsequently, we have the following inequality:

$$\|\mathbf{e}_T\|_\infty \leq \sum_{i=0}^{T-1} \|\mathbf{A}\|_\infty^i \|\varepsilon_{k+T-i}\|_\infty \leq \varepsilon_{\max} \sum_{i=0}^{T-1} \|\mathbf{A}\|_\infty^i. \tag{22}$$

It is noteworthy that, in this proof, we assume the one-step prediction error $\varepsilon_{\max}$ is bounded. This assumption is mild, since the objective function of our QP-based mapping is explicitly designed to minimize the one-step prediction error.

Furthermore, by the geometric-series formula, if $\|\mathbf{A}\|_\infty < 1$, we have

$$\|\mathbf{e}_T\|_\infty \leq \varepsilon_{\max} \sum_{i=0}^{T-1} \|\mathbf{A}\|_\infty^i = \varepsilon_{\max} \frac{1 - \|\mathbf{A}\|_\infty^T}{1 - \|\mathbf{A}\|_\infty} \leq \frac{\varepsilon_{\max}}{1 - \|\mathbf{A}\|_\infty}, \qquad \forall T \geq 1. \tag{23}$$

Therefore, if $\|\mathbf{A}\|_\infty < 1$, the cumulative prediction error over any prediction horizon $T$ is bounded. This completes the proof. $\qquad\square$

## C. Proof of Proposition 4.2

**Proposition C.1** (Restatement of Proposition 4.2). *The QP problem in (9) is equivalent to the $N$ independent sub-QP problems in (11).*

*Proof.* We first give the following definition:

$$\mathbf{J} := \mathbf{Z}(\boldsymbol{\theta})\mathbf{A}^{\mathrm{T}} + \mathbf{U}\mathbf{B}^{\mathrm{T}} - \mathbf{Z}^+(\boldsymbol{\theta}) \in \mathbb{R}^{N_{\mathrm{QP}} \times N}. \tag{24}$$

Then, the first term in (9) can be written as

$$\left\|\mathbf{Z}(\boldsymbol{\theta})\mathbf{A}^{\mathrm{T}} + \mathbf{U}\mathbf{B}^{\mathrm{T}} - \mathbf{Z}^+(\boldsymbol{\theta})\right\|_{\mathrm{F}}^2 = \|\mathbf{J}\|_{\mathrm{F}}^2. \tag{25}$$

Next, we decompose $\|\mathbf{J}\|_{\mathrm{F}}^2$ by columns. Specifically, we denote $\mathbf{J}^{(h)} \in \mathbb{R}^{N_{\mathrm{QP}}}$ as the $h$-th column of $\mathbf{J}$. Then, the following equalities hold:

$$\|\mathbf{J}\|_{\mathrm{F}}^2 = \sum_{h=1}^N \left\|\mathbf{J}^{(h)}\right\|_2^2. \tag{26}$$

$$\mathbf{J}^{(h)} = \left(\mathbf{Z}(\boldsymbol{\theta})\mathbf{A}^{\mathrm{T}}\right)^{(h)} + \left(\mathbf{U}\mathbf{B}^{\mathrm{T}}\right)^{(h)} - \mathbf{Z}^{+(h)}(\boldsymbol{\theta}). \tag{27}$$

Since the $h$-th column of $\mathbf{Z}(\boldsymbol{\theta})\mathbf{A}^{\mathrm{T}}$ is $\mathbf{Z}(\boldsymbol{\theta})$ multiplied by the $h$-th column of $\mathbf{A}^{\mathrm{T}}$, we have

$$\left(\mathbf{Z}(\boldsymbol{\theta})\mathbf{A}^{\mathrm{T}}\right)^{(h)} = \mathbf{Z}(\boldsymbol{\theta})\mathbf{A}^{\mathrm{T}(h)}. \tag{28}$$

Similarly, we have

$$\left(\mathbf{U}\mathbf{B}^{\mathrm{T}}\right)^{(h)} = \mathbf{U}\mathbf{B}^{\mathrm{T}(h)}. \tag{29}$$

By combining (27)–(29), we obtain

$$\mathbf{J}^{(h)} = \mathbf{Z}(\boldsymbol{\theta})\mathbf{A}^{\mathrm{T}(h)} + \mathbf{U}\mathbf{B}^{\mathrm{T}(h)} - \mathbf{Z}^{+(h)}(\boldsymbol{\theta}). \tag{30}$$

Subsequently, substituting (30) into (26) yields

$$\|\mathbf{J}\|_{\mathrm{F}}^2 = \sum_{h=1}^N \left\|\mathbf{Z}(\boldsymbol{\theta})\mathbf{A}^{\mathrm{T}(h)} + \mathbf{U}\mathbf{B}^{\mathrm{T}(h)} - \mathbf{Z}^{+(h)}(\boldsymbol{\theta})\right\|_2^2. \tag{31}$$

Moreover, the slack-penalty term in (9) satisfies

$$\alpha \sum_{i=1}^N \xi_i^2 = \sum_{h=1}^N \alpha \xi_h^2. \tag{32}$$

Next, we consider the constraints in (9). By replacing the index $i$ with $h$ in the first set of constraints (i.e., the slack constraints) in (9), we obtain

$$\sum_{j=1}^N r_{h,j} + \xi_h \leq 1 - c, \qquad h = 1, \ldots, N. \tag{33}$$

Similarly, the second set of constraints (i.e., the box constraints) in (9) can be rewritten as

$$-r_{h,j} \le a_{h,j} \le r_{h,j}, \qquad h = 1, \ldots, N, \ j = 1, \ldots, N, \tag{34}$$

By combining (31)-(34), the QP problem in (9) can be equivalently rewritten as

$$
\begin{aligned}
\underset{\mathbf{A},\mathbf{B},r_{h,j},\xi_h}{\arg\min} \quad & \sum_{h=1}^{N} \left( \left\| \mathbf{Z}(\boldsymbol{\theta})\mathbf{A}^{\mathrm{T}(h)} + \mathbf{U}\mathbf{B}^{\mathrm{T}(h)} - \mathbf{Z}^{+(h)}(\boldsymbol{\theta}) \right\|_2^2 + \alpha \xi_h^2 \right) \\
\text{s.t.} \quad & \sum_{j=1}^{N} r_{h,j} + \xi_h \le 1 - c, \qquad h = 1, \ldots, N, \\
& -r_{h,j} \le a_{h,j} \le r_{h,j}, \qquad h = 1, \ldots, N, \ j = 1, \ldots, N.
\end{aligned}
\tag{35}
$$

From (35), one can see that for a fixed $h$, the $h$-th term in the objective function and the constraints with index $h$ only involve decision variables $\mathbf{A}^{\mathrm{T}(h)}$, $\mathbf{B}^{\mathrm{T}(h)}$, $\xi_h$, and $\{r_{h,j}\}_{j=1}^N$, and do not involve decision variables indexed by $\tilde{h} \ne h$. Therefore, the QP problem in (35) can be decomposed into $N$ independent sub-QP problems indexed by $h$, i.e.,

$$
\begin{aligned}
\underset{\mathbf{A}^{\mathrm{T}(h)},\mathbf{B}^{\mathrm{T}(h)},r_{h,j},\xi_h}{\min} \quad & \left\| \mathbf{Z}(\boldsymbol{\theta})\mathbf{A}^{\mathrm{T}(h)} + \mathbf{U}\mathbf{B}^{\mathrm{T}(h)} - \mathbf{Z}^{+(h)}(\boldsymbol{\theta}) \right\|_2^2 + \alpha \xi_h^2 \\
\text{s.t.} \quad & \sum_{j=1}^{N} r_{h,j} + \xi_h \le 1 - c, \\
& -r_{h,j} \le a_{h,j} \le r_{h,j}, \quad j = 1, \ldots, N,
\end{aligned}
\tag{36}
$$

and minimizing (35) is equivalent to solving these $N$ sub-QP problems independently and assembling their solutions. This completes the proof. $\square$

## D. End-to-End Training Procedure of QPKO

The end-to-end training procedure of QPKO is summarized in Algorithm 1, where $Epoch_{max}$ denotes the maximum number of training epochs and $N_{\mathrm{Tra}}$ denotes the number of trajectories in the trajectory dataset $\mathcal{D}_{\mathrm{Tra}}$.

---

**Algorithm 1** End-to-End Training of QPKO

---

1: **Input:** operational trajectories of the modeled system, hyperparameters of QPKO
2: Construct QP dataset $\mathcal{D}_{\mathrm{QP}} = \{\mathbf{x}_1^{1:N_{\mathrm{QP}}}, \mathbf{x}_2^{1:N_{\mathrm{QP}}}, \mathbf{u}_1^{1:N_{\mathrm{QP}}}\}$ and trajectory dataset $\mathcal{D}_{\mathrm{Tra}} = \{\mathbf{x}_{1:T+1}^{1:N_{\mathrm{Tra}}}, \mathbf{u}_{1:T}^{1:N_{\mathrm{Tra}}}\}$
3: Initialize $\boldsymbol{\theta}$
4: **for** $epoch = 1, \ldots, Epoch_{max}$ **do**
5:      Sample a mini-batch from $\mathcal{D}_{\mathrm{Tra}}$ with batch size $n_{\mathrm{Tra}}$, i.e., $\{\mathbf{x}_{1:T+1}^{1:n_{\mathrm{Tra}}}, \mathbf{u}_{1:T}^{1:n_{\mathrm{Tra}}}\}$
6:      Compute $\mathbf{z}_{1:T+1}^{1:n_{\mathrm{Tra}}}$ from $\mathbf{x}_{1:T+1}^{1:n_{\mathrm{Tra}}}$ using the observable module (5)
7:      Compute $\mathbf{A}^*(\boldsymbol{\theta})$ and $\mathbf{B}^*(\boldsymbol{\theta})$ based on $\mathcal{D}_{\mathrm{QP}}$ and $\boldsymbol{\theta}$ via the QP-based mapping module (13)
8:      Compute $\hat{\mathbf{z}}_{2:T+1}^{1:n_{\mathrm{Tra}}}$ from $\mathbf{z}_1^{1:n_{\mathrm{Tra}}}$, $\mathbf{u}_{1:T}^{1:n_{\mathrm{Tra}}}$, $\mathbf{A}^*(\boldsymbol{\theta})$ and $\mathbf{B}^*(\boldsymbol{\theta})$ via the linear recursion module (14)
9:      Evaluate $\mathcal{L}$ based on $\hat{\mathbf{z}}_{2:T+1}^{1:n_{\mathrm{Tra}}}$ and $\mathbf{z}_{2:T+1}^{1:n_{\mathrm{Tra}}}$ via (15)
10:      Update $\boldsymbol{\theta}$ via Adam (Kingma & Ba, 2014) optimizer
11: **end for**

---

As shown in the above algorithm, the training of QPKO uses two datasets, i.e., $\mathcal{D}_{\mathrm{QP}}$ and $\mathcal{D}_{\mathrm{Tra}}$. They are constructed from the same operational trajectories of the modeled dynamical system. The difference between them lies only in how these trajectories are reorganized.

Specifically, in $\mathcal{D}_{\mathrm{QP}}$, the trajectories are reorganized into $N_{\mathrm{QP}}$ one-step transition tuples, i.e., $\mathcal{D}_{\mathrm{QP}} = \{\mathbf{x}_1^{1:N_{\mathrm{QP}}}, \mathbf{x}_2^{1:N_{\mathrm{QP}}}, \mathbf{u}_1^{1:N_{\mathrm{QP}}}\}$. Conversely, in $\mathcal{D}_{\mathrm{Tra}}$, the trajectories are reorganized into $N_{\mathrm{Tra}}$ trajectory segments, each containing $T + 1$ consecutive states and $T$ corresponding inputs, i.e., $\mathcal{D}_{\mathrm{Tra}} = \{\mathbf{x}_{1:T+1}^{1:N_{\mathrm{Tra}}}, \mathbf{u}_{1:T}^{1:N_{\mathrm{Tra}}}\}$.

# E. Experimental Details

## E.1. Four Nonlinear Dynamical Systems and Data Collection

### (1) CartPole

The dynamics of CartPole can be expressed as (Sakib & Pan, 2025)

$$
\begin{aligned}
\dot{x}_1 &= x_3, \\
\dot{x}_2 &= x_4, \\
\dot{x}_3 &= \frac{u + m_{\mathrm{p}} \sin x_2 \left( l x_4^2 - g \cos x_2 \right)}{m_{\mathrm{c}} + m_{\mathrm{p}} \sin^2 x_2}, \\
\dot{x}_4 &= \frac{u \cos x_2 + m_{\mathrm{p}} l x_4^2 \cos x_2 \sin x_2 - (m_{\mathrm{c}} + m_{\mathrm{p}}) g \sin x_2}{l \left( m_{\mathrm{c}} + m_{\mathrm{p}} \sin^2 x_2 \right)}.
\end{aligned}
\tag{37}
$$

Here, $x_1$ denotes the cart position, $x_2$ denotes the pole angle, $x_3$ denotes the cart velocity, $x_4$ denotes the angular velocity of the pole, and $u$ denotes the external force input applied on the cart. As for the parameters of this system, $m_{\mathrm{c}} = 4$ (kg), $m_{\mathrm{p}} = 1$ (kg), $l = 1$ (m), $g = 9.81$ (m/s$^2$). Meanwhile, the fourth-order Runge–Kutta method is adopted to simulate this system with a discrete time step of $\Delta t = 0.05$ s.

As for the data collection on CartPole, we generate 100 trajectories with a length of 200 time steps. Following (Sakib & Pan, 2025), the initial state is sampled as $x_1(1) \sim \mathcal{U}[-1, 1]$, $x_2(1) \sim \mathcal{U}\left[-\frac{\pi}{2}, \frac{\pi}{2}\right]$, $x_3(1) \sim \mathcal{U}[-0.1, 0.1]$, and $x_4(1) \sim \mathcal{U}[-0.1, 0.1]$. Here, $x_i(1)$ denotes the initial value of $x_i$. $\mathcal{U}[a, b]$ denotes the uniform distribution over the interval $[a, b]$. The input is generated using exponentially decaying sine waves, i.e.

$$
u(t) = A \, e^{-\lambda t} \sin(\omega t + \phi),
\tag{38}
$$

where the amplitude, frequency, phase, and decay rate are randomly sampled as $A \sim \mathcal{U}[0.5, 2.0]$, $\omega \sim \mathcal{U}[0.2, 2.7]$, $\phi \sim \mathcal{U}[0, 2\pi]$, and $\lambda \sim \mathcal{U}[0.02, 0.07]$. To collect discrete input, we sample the continuous-time input signal in (38) as $u_k = u(k \Delta t)$.

### (2) Forced Van der Pol oscillator (VDP)

The VDP is governed by the following dynamics:

$$
\begin{aligned}
\dot{x}_1 &= 2x_2, \\
\dot{x}_2 &= -0.8x_1 + 2x_2 - 10x_1^2 x_2 - u.
\end{aligned}
\tag{39}
$$

Here, $x_1$ and $x_2$ denote the two system states and $u$ denotes the external input. We adopt the fourth-order Runge-Kutta method to simulate this system with a discrete time step of $\Delta t = 0.01$ s.

As for the data collection on VDP, we generate 100 trajectories with a length of 200 time steps. Following (Korda & Mezic, 2018), the initial state is sampled as $x_1(1) \sim \mathcal{U}[-1, 1]$ and $x_2(1) \sim \mathcal{U}[-1, 1]$. The input is sampled as $u_k \sim \mathcal{U}[-1, 1]$.

### (3) Wind Turbine (WT).
In this work, we focus on the electromechanical transients of the NREL 5MW WT (Jonkman et al., 2009), mainly including the dynamics of drivetrain and pitch actuator. These dynamics can be expressed as

$$
\begin{aligned}
\dot{\omega}_{\mathrm{r}} &= \frac{1}{J_{\mathrm{t}}} \left( \frac{\frac{1}{2}\pi \rho R^2 v^3 \, C_{\mathrm{p}}(\lambda, \beta)}{\omega_{\mathrm{r}}} - \frac{P_{\mathrm{e}}^{\mathrm{ref}}}{\omega_{\mathrm{r}}} \right), \\
\dot{\beta} &= \frac{1}{T_{\mathrm{p}}} \left( \beta^{\mathrm{ref}} - \beta \right).
\end{aligned}
\tag{40}
$$

Here $\omega_{\mathrm{r}}$ denotes the rotor speed, $\beta$ denotes the pitch angle, $v$ denotes the wind speed, $P_{\mathrm{e}}^{\mathrm{ref}}$ denotes active power reference, $\beta^{\mathrm{ref}}$ denotes pitch angle reference, $J_{\mathrm{t}}$ is the combined inertia of the rotor and the generator, $T_{\mathrm{p}}$ is the time constant of the pitch actuator, $\rho$ is the air density, $C_{\mathrm{p}}(\lambda, \beta)$ is the power coefficient, which is a nonlinear function of the tip-speed ratio $\lambda$ and the pitch angle $\beta$. The tip-speed ratio is defined as $\lambda = \omega_{\mathrm{r}} R / v$. The state of the WT can be expressed as $[\omega_{\mathrm{r}}, \beta]$, while the input can be expressed as $[P_{\mathrm{e}}^{\mathrm{ref}}, \beta^{\mathrm{ref}}, v]$. The detailed parameters of the NREL 5MW WT can be found in (Tian et al., 2025). We simulate the WT system using the forward Euler method. To ensure numerical accuracy, the integration time step is set to $\Delta t = 0.01$ s, while the data in operational trajectories are sampled every 0.1 s.

Then, we detail the data collection on WT. We generate 100 trajectories with a length of 200 time steps. The input, i.e., $[P_{\mathrm{e}}^{\mathrm{ref}}, \beta^{\mathrm{ref}}, v]$, is first collected from a real-world wind farm and then scaled to match the operating range of the NREL 5MW WT. The initial state, i.e., $[\omega_{\mathrm{r}}(1), \beta(1)]$, is randomly sampled. Specifically, $\omega_{\mathrm{r}}(1) \sim \mathcal{U}[0.9, 1.25]$ (rad/s), and $\beta(1) \sim \mathcal{U}[0, 10]$ (deg).

Moreover, according to (40), the nonlinearity arises only in the rotor speed dynamic, while the pitch angle dynamic is linear. Accordingly, we conduct Koopman operator theory-based modeling only for the rotor speed dynamic and retain the pitch actuator model in its original linear form.

**(4) Tethered Space Robot (TSR)**

The dynamics of TSR can be expressed as (Jin et al., 2024):

$$
\begin{aligned}
\dot{x}_1 &= x_2, \\
\dot{x}_2 &= -2\frac{x_4}{x_3 + 1}(x_2 + 1) \ - \ 3\sin(x_1)\cos(x_1), \\
\dot{x}_3 &= x_4, \\
\dot{x}_4 &= (x_3 + 1)\Big((x_2 + 1)^2 + 3\cos^2(x_1) - 1\Big) \ - \ u.
\end{aligned}
\tag{41}
$$

Here, $\dot{(\cdot)}$ denotes the derivative with respect to a dimensionless time $\tau$. $x_1$ is the in-plane angle, $x_2$ is the in-plane angular velocity, $x_3 = l - 1$, $l$ is the tether deployment ratio, $x_4$ is the tether deployment rate. The control input $u$ is the dimensionless tether tension. The fourth-order Runge-Kutta method is adopted to simulate TSR with a dimensionless discrete time step of $\Delta\tau = 0.01$ (a.u.).

As for the data collection on TSR, we generate 100 trajectories with a length of 200 time steps. Following (Jin et al., 2024), the initial state is randomly sampled. Specifically, $x_1(1) \sim \mathcal{U}[-2.5, 0.5]$, $x_2(1) \sim \mathcal{U}[-1, 1]$, $x_3(1) \sim \mathcal{U}[-0.99, 0]$, and $x_4(1) \sim \mathcal{U}[0, 2]$. Following (Jin et al., 2025a), the input is generated using a predetermined control law, i.e.,

$$
u_k = 4x_{3,k} + 2x_{4,k} + 3.
\tag{42}
$$

**E.2. Hyperparameters**

The hyperparameters of QPKO, DKO, EDMD-DL, and NDMD are, respectively, listed in Table 4, Table 5, Table 6, and Table 7. The training of the four frameworks all follows a curriculum learning strategy (Sakib & Pan, 2025). Specifically, the prediction horizon for loss computation is initialized to 10 and increases by 15 every $0.1Epoch_{\max}$, until it reaches 85 at $0.5Epoch_{\max}$. After $0.5Epoch_{\max}$, this horizon is fixed at 85. The hyperparameters are selected via grid search based on validation performance. To facilitate a fair comparison of training efficiency, we further tune the hyperparameters of the three baselines to speed up training while preserving validation performance.

*Table 4.* Hyperparameters of QPKO.

| HYPERPARAMETERS | VALUE |
|---|---|
| BATCH SIZE | FULL BATCH |
| LEARNING RATE (CARTPOLE, VDP) | $1.0 \times 10^{-3}$ |
| LEARNING RATE (WT) | $1.0 \times 10^{-5}$ |
| LEARNING RATE (TSR) | $7.5 \times 10^{-4}$ |
| MAXIMUM TRAINING EPOCH $Epoch_{\max}$ | $5.0 \times 10^{3}$ |
| PREDICTION HORIZON FOR LOSS COMPUTATION | $10 \to 25 \to 40 \to 55 \to 70 \to 85$ |
| STRUCTURE OF THE MLP FOR STATE LIFTING (CARTPOLE, TSR) | $(4, 128, 128, 128, 128, 100)$ |
| STRUCTURE OF THE MLP FOR STATE LIFTING (VDP, WT) | $(2, 128, 128, 128, 128, 100)$ |
| ACTIVATION FUNCTION OF THE MLP FOR STATE LIFTING | TANH |
| DIMENSION OF LIFTED STATE (CARTPOLE, TSR) | 104 |
| DIMENSION OF LIFTED STATE (VDP) | 102 |
| DIMENSION OF LIFTED STATE (WT) | 101 |
| PENALTY WEIGHT OF THE CONSTRAINT SLACK IN QP-BASED MAPPING MODULE | $1.0 \times 10^{-2}$ |

*Table 5.* Hyperparameters of DKO.

| HYPERPARAMETERS | VALUE |
|---|---|
| BATCH SIZE | 2048 |
| LEARNING RATE (CARTPOLE, VDP, TSR) | $1.0 \times 10^{-4}$ |
| LEARNING RATE (WT) | $1.0 \times 10^{-5}$ |
| MAXIMUM TRAINING EPOCH $Epoch_{\max}$ (CARTPOLE, TSR) | $2.0 \times 10^{4}$ |
| MAXIMUM TRAINING EPOCH $Epoch_{\max}$ (VDP) | $3.0 \times 10^{4}$ |
| MAXIMUM TRAINING EPOCH $Epoch_{\max}$ (WT) | $5.0 \times 10^{3}$ |
| PREDICTION HORIZON FOR LOSS COMPUTATION | $10 \rightarrow 25 \rightarrow 40 \rightarrow 55 \rightarrow 70 \rightarrow 85$ |
| STRUCTURE OF THE MLP FOR STATE LIFTING (CARTPOLE, TSR) | $(4, 128, 128, 128, 128, 100)$ |
| STRUCTURE OF THE MLP FOR STATE LIFTING (VDP, WT) | $(2, 128, 128, 128, 128, 100)$ |
| ACTIVATION FUNCTION OF THE MLP FOR STATE LIFTING | TANH |
| DIMENSION OF LIFTED STATE (CARTPOLE, TSR) | 104 |
| DIMENSION OF LIFTED STATE (VDP) | 102 |
| DIMENSION OF LIFTED STATE (WT) | 101 |

*Table 6.* Hyperparameters of EDMD-DL.

| HYPERPARAMETERS | VALUE |
|---|---|
| BATCH SIZE | 2048 (FULL BATCH FOR LS SOLVING) |
| LEARNING RATE (CARTPOLE) | $1.0 \times 10^{-4}$ |
| LEARNING RATE (VDP, TSR) | $2.5 \times 10^{-4}$ |
| LEARNING RATE (WT) | $3.0 \times 10^{-6}$ |
| MAXIMUM TRAINING EPOCH $Epoch_{\max}$ (CARTPOLE) | $4.0 \times 10^{4}$ |
| MAXIMUM TRAINING EPOCH $Epoch_{\max}$ (VDP) | $1.5 \times 10^{4}$ |
| MAXIMUM TRAINING EPOCH $Epoch_{\max}$ (WT) | $5.0 \times 10^{3}$ |
| MAXIMUM TRAINING EPOCH $Epoch_{\max}$ (TSR) | $1.0 \times 10^{4}$ |
| PREDICTION HORIZON FOR LOSS COMPUTATION | $10 \rightarrow 25 \rightarrow 40 \rightarrow 55 \rightarrow 70 \rightarrow 85$ |
| STRUCTURE OF THE MLP FOR STATE LIFTING (CARTPOLE, TSR) | $(4, 128, 128, 128, 128, 100)$ |
| STRUCTURE OF THE MLP FOR STATE LIFTING (VDP, WT) | $(2, 128, 128, 128, 128, 100)$ |
| ACTIVATION FUNCTION OF THE MLP FOR STATE LIFTING | TANH |
| DIMENSION OF LIFTED STATE (CARTPOLE, TSR) | 104 |
| DIMENSION OF LIFTED STATE (VDP) | 102 |
| DIMENSION OF LIFTED STATE (WT) | 101 |

*Table 7.* Hyperparameters of NDMD.

| HYPERPARAMETERS | VALUE |
|---|---|
| BATCH SIZE | 2048 (FULL BATCH FOR LS SOLVING) |
| LEARNING RATE (CARTPOLE, VDP) | $1.0 \times 10^{-4}$ |
| LEARNING RATE (WT) | $1.0 \times 10^{-5}$ |
| LEARNING RATE (TSR) | $2.5 \times 10^{-4}$ |
| MAXIMUM TRAINING EPOCH $Epoch_{\max}$ (CARTPOLE) | $1.0 \times 10^{4}$ |
| MAXIMUM TRAINING EPOCH $Epoch_{\max}$ (VDP) | $1.0 \times 10^{4}$ |
| MAXIMUM TRAINING EPOCH $Epoch_{\max}$ (WT) | $5.0 \times 10^{3}$ |
| MAXIMUM TRAINING EPOCH $Epoch_{\max}$ (TSR) | $5.0 \times 10^{3}$ |
| PREDICTION HORIZON FOR LOSS COMPUTATION | $10 \rightarrow 25 \rightarrow 40 \rightarrow 55 \rightarrow 70 \rightarrow 85$ |
| STRUCTURE OF THE MLP FOR STATE LIFTING (CARTPOLE, TSR) | $(4, 128, 128, 128, 128, 100)$ |
| STRUCTURE OF THE MLP FOR STATE LIFTING (VDP, WT) | $(2, 128, 128, 128, 128, 100)$ |
| ACTIVATION FUNCTION OF THE MLP FOR STATE LIFTING | TANH |
| DIMENSION OF LIFTED STATE (CARTPOLE, TSR) | 104 |
| DIMENSION OF LIFTED STATE (VDP) | 102 |
| DIMENSION OF LIFTED STATE (WT) | 101 |

### E.3. Formulation of NMAE

The NMAE for predictive horizon $T$ can be expressed as

$$NMAE = \frac{1}{N_{\text{pred}} \cdot T \cdot n} \sum_{j=1}^{N_{\text{pred}}} \sum_{k=2}^{T+1} \sum_{i=1}^{n} \frac{\left|\hat{x}_{i,k}^j - x_{i,k}^j\right|}{x_{i,\max} - x_{i,\min}}. \tag{43}$$

Here, $N_{\text{pred}}$ denotes the number of prediction trajectories. $\hat{x}_{i,k}^j$ and $x_{i,k}^j$, respectively, denote the predicted value and true value of the $i$-th system state at time step $k$ in the $j$-th prediction trajectory. $x_{i,\max}$ and $x_{i,\min}$, respectively, denote the maximum and minimum values of the $i$-th system state in the collected data set.

### E.4. Formulation of the Per-Step Prediction Error

The per-step prediction error at the time step $k$, i.e., $E_{\text{per},k}$, can be expressed as

$$E_{\text{per},k} = \frac{1}{N_{\text{pred}} \cdot n} \sum_{j=1}^{N_{\text{pred}}} \sum_{i=1}^{n} \frac{\left|\hat{x}_{i,k}^j - x_{i,k}^j\right|}{x_{i,\max} - x_{i,\min}}. \tag{44}$$

### E.5. Details of MPC Implementation

#### (1) Formulation of MPC

There are three key elements in MPC problem formulation, that is, (i) the prediction model, (ii) objective function, and (iii) constraints. In this work, the prediction model is given by Koopman operator theory-based modeling. Details about using the Koopman model to implement MPC are given in (Korda & Mezic, 2018). The objective function and constraints considered in this work are as follows:

$$\min_{\{\mathbf{u}_{k+h-1|k}\}_{h=1}^T} \sum_{h=1}^{T} \sum_{i=1}^{n} q_i \left(x_{i,k+h|k} - x_{i,k+h|k}^{\text{ref}}\right) \tag{45}$$
$$\text{s.t.} \quad \mathbf{u}_{\min} \leq \mathbf{u}_{k+h-1|k} \leq \mathbf{u}_{\max}, \quad h = 1, \ldots, T-1.$$

Here, $q_i$ is the weight of the $i$-th system state's tracking error. The subscript "$k+h-1|k$" denotes the prediction or decision at time step $k+h-1$ computed based on information available at time step $k$. $\mathbf{u}_{\max}$ and $\mathbf{u}_{\min}$, respectively, denote the maximum and minimum values of the input. The superscript "ref" denotes the reference value.

As for the settings in MPC, in each framework and nonlinear system, the prediction horizon $T$ in MPC is set to 85 and the $\{q_i\}_{i=1}^n$ is set to 1. As for the $\mathbf{u}_{\min}$ and $\mathbf{u}_{\max}$, on CartPole, they are, respectively, set to -20 and 20. On VDP and TSR, the input ranges are, respectively, [-1, 1] and [0, 5]. On WT, there are three inputs, i.e., active power reference $P_e^{\text{ref}}$, pitch angle reference $\beta_{\text{ref}}$, and wind speed $v$. Among them, $P_e^{\text{ref}}$ and $\beta_{\text{ref}}$ are decision variables of MPC. Their ranges are, respectively, set to $[0, 5 \times 10^6]$ and [0, 10]. The wind speed is determined by the external environment.

#### (2) Random sampling of initial states or tracking references

On CartPole, VDP, and TSR, the control objective is driving the system from a randomly sampled initial state to the origin. For each system, 10 initial states are randomly sampled. Specifically, on CartPole, the initial values of $x_1$-$x_4$ are all sampled from $\mathcal{U}[-0.5, 0.5]$. On VDP, the initial values of $x_1$ and $x_2$ are both sampled from $\mathcal{U}[-0.5, 0.5]$. On TSR, $x_1$ and $x_2$ are initialized to 0, while the initial values of $x_3$ and $x_4$ are, respectively, sampled from $\mathcal{U}[-0.99, 0]$ and $\mathcal{U}[-0.5, 0.5]$.

On WT, MPC coordinates the active power reference $P_e^{\text{ref}}$ and pitch angle reference $\beta_{\text{ref}}$ under the time-varying wind speed $v$ to make the rotor speed $\omega_r$ track a sinusoidal reference with a period of 20 s and randomly sampled parameters. Here, 10 sinusoidal references are sampled. Specifically, the maximum and minimum values of these references are sampled from $\mathcal{U}[0.98, 1.02]$ and $\mathcal{U}[1.23, 1.27]$.

## E.6. Control Performance Metrics

The instantaneous setpoint error at time step $k$, i.e., $E_{\text{ins},k}$, is formulated as

$$E_{\text{ins},k} = \frac{1}{n} \sum_{i=1}^{n} |x_{i,k} - 0| \, . \tag{46}$$

The cumulative tracking error at time step $k$, i.e., $E_{\text{cum},k}$, is formulated as

$$E_{\text{cum},k} = \frac{1}{n} \sum_{t=1}^{k} \sum_{i=1}^{n} \left| x_{i,t} - x_{i,t}^{\text{ref}} \right| \, . \tag{47}$$

# F. Ablation Study on the Multi-Step Accuracy-Oriented Constraints

To evaluate the effect of the multi-step accuracy-oriented constraints (8) on QPKO, we conduct an ablation study. Table 8 reports the NMAE of QPKO with and without these constraints. From this table, one can see that the QPKO with these constraints consistently yields lower NMAE across all four systems and horizons. This demonstrates the effectiveness of these constraints in improving the modeling accuracy. One potential reason for this improvement is as follows: These constraints make the QP-based mapping module account for multi-step prediction accuracy when computing the global linear model. Consequently, the objective of the QP-based mapping module better aligns with the QPKO's overall training goal, i.e., minimizing the multi-step rollout error.

*Table 8.* The NMAE given by QPKO with and without the multi-step accuracy-oriented constraints (8) at prediction horizons $T \in \{50, 100, 150, 200\}$. Results are reported as mean±std (in units of $10^{-2}$). The mean and standard deviation are computed over 5 independent training trials. The best-performing results are highlighted in **bold**.

| | CARTPOLE ($\times 10^{-2}$) | | | | VDP ($\times 10^{-2}$) | | | |
|---|---|---|---|---|---|---|---|---|
| VARIANTS | $T = 50$ | $T = 100$ | $T = 150$ | $T = 200$ | $T = 50$ | $T = 100$ | $T = 150$ | $T = 200$ |
| QPKO (w/o (8)) | 0.335±0.054 | 0.689±0.107 | 1.139±0.213 | 1.882±0.345 | 0.135±0.020 | 0.565±0.286 | 1.453±0.691 | 2.306±0.796 |
| QPKO (w/ (8)) | **0.325**±**0.057** | **0.663**±**0.096** | **1.073**±**0.165** | **1.783**±**0.257** | **0.111**±**0.019** | **0.330**±**0.082** | **0.885**±**0.173** | **1.640**±**0.375** |

| | WT ($\times 10^{-2}$) | | | | TSR ($\times 10^{-2}$) | | | |
|---|---|---|---|---|---|---|---|---|
| VARIANTS | $T = 50$ | $T = 100$ | $T = 150$ | $T = 200$ | $T = 50$ | $T = 100$ | $T = 150$ | $T = 200$ |
| QPKO (w/o (8)) | 0.518±0.109 | 0.893±0.149 | 1.274±0.229 | 1.665±0.327 | 0.224±0.070 | 0.306±0.065 | 0.719±0.083 | 1.197±0.159 |
| QPKO (w/ (8)) | **0.506**±**0.090** | **0.841**±**0.188** | **1.174**±**0.277** | **1.520**±**0.347** | **0.142**±**0.056** | **0.255**±**0.068** | **0.649**±**0.165** | **1.147**±**0.347** |

# G. Additional Modeling Accuracy Results

## G.1. Modeling Accuracy under Measurement Noise

In this section, CartPole is selected as the representative system to evaluate the modeling accuracy of different frameworks under measurement noise.

To this end, we construct a noisy dataset by incorporating uniformly distributed noise into the system states of the noise-free CartPole operational dataset, that is

$$\tilde{x}_{i,k} = x_{i,k}(1 + \epsilon_{i,k}), \quad \epsilon_{i,k} \sim \mathcal{U}\left[-p, p\right] . \tag{48}$$

Here, $\tilde{x}_{i,k}$ denotes the noisy measurement of the $i$-th system state at time step $k$. $\epsilon_{i,k}$ is the relative noise term. It is independently sampled for each system state and time step. $p$ denotes the noise level. The training and validation are conducted on the noisy dataset. When testing the modeling accuracy, the noisy measurements of system states are used for state lifting, while the corresponding noise-free trajectories are adopted as the ground truth for computing prediction errors.

The modeling accuracy of each framework is evaluated under two noise levels, i.e., $p = 3\%$ and $p = 6\%$. The corresponding NMAEs for multi-step predictions are reported in Table 9. According to this table, QPKO achieves the lowest NMAE across

all noise levels and prediction horizons. Specifically, under $p = 3\%$, QPKO reduces the mean NMAE by 21.7%, 23.0%, 30.0%, and 26.5% at $T = 50, 100, 150$, and 200. Under $p = 6\%$, the corresponding reductions are, respectively, 30.2%, 32.5%, 36.8%, and 33.4%. These results indicate that QPKO exhibits superior modeling accuracy under measurement noise, and this superiority is more pronounced under the higher noise level, i.e., $p = 6\%$.

*Table 9.* The NMAE given by each framework on CartPole under noise levels $p \in \{3\%, 6\%\}$ and prediction horizons $T \in \{50, 100, 150, 200\}$. Results are reported as mean±std (in units of $10^{-2}$). The mean and standard deviation are computed over 5 independent training trials. The best-performing results are highlighted in **bold**, while the second-best results are underlined.

| | $p = 3\%$ | | | | $p = 6\%$ | | | |
|---|---|---|---|---|---|---|---|---|
| FRAMEWORK | $T = 50$ | $T = 100$ | $T = 150$ | $T = 200$ | $T = 50$ | $T = 100$ | $T = 150$ | $T = 200$ |
| DKO | 0.550±0.095 | 1.114±0.162 | 2.147±0.257 | 3.431±0.337 | 0.560±0.101 | 1.156±0.156 | 2.217±0.286 | 3.503±0.431 |
| EDMD-DL | 0.486±0.096 | 1.239±0.244 | 2.340±0.400 | 3.520±0.540 | 0.878±0.094 | 2.021±0.276 | 3.473±0.507 | 4.921±0.767 |
| NDMD | 0.452±0.088 | 0.930±0.154 | 1.744±0.309 | 2.868±0.479 | 0.612±0.102 | 1.116±0.161 | 2.079±0.271 | 3.407±0.370 |
| QPKO | **0.354±0.048** | **0.716±0.077** | **1.220±0.160** | **2.107±0.271** | **0.391±0.055** | **0.753±0.075** | **1.313±0.149** | **2.268±0.319** |

Moreover, we also evaluate the effectiveness of the multi-step accuracy-oriented constraints (8) on modeling accuracy under measurement noise. The NMAE given by QPKO with and without constraints (8) is reported in Table 10. As shown in this table, these constraints effectively reduce the NMAE across all noise levels and prediction horizons, indicating their effectiveness in improving modeling accuracy under measurement noise. Meanwhile, compared to the lower noise level $p = 3\%$, this effectiveness is more obvious under the higher noise level $p = 6\%$.

*Table 10.* The NMAE given by QPKO on CartPole with and without the multi-step accuracy-oriented constraints (8) under noise levels $p \in \{3\%, 6\%\}$ and prediction horizons $T \in \{50, 100, 150, 200\}$. Results are reported as mean±std (in units of $10^{-2}$). The mean and standard deviation are computed over 5 independent training trials. The best-performing results are highlighted in **bold**.

| | $p = 3\%$ | | | | $p = 6\%$ | | | |
|---|---|---|---|---|---|---|---|---|
| VARIANTS | $T = 50$ | $T = 100$ | $T = 150$ | $T = 200$ | $T = 50$ | $T = 100$ | $T = 150$ | $T = 200$ |
| QPKO (w/o (8)) | 0.357±0.051 | 0.724±0.093 | 1.299±0.192 | 2.296±0.341 | 0.408±0.067 | 0.801±0.113 | 1.539±0.277 | 2.822±0.530 |
| QPKO (w/ (8)) | **0.354±0.048** | **0.716±0.077** | **1.220±0.160** | **2.107±0.271** | **0.391±0.055** | **0.753±0.075** | **1.313±0.149** | **2.268±0.319** |

### G.2. Sensitivity of QPKO's Modeling Accuracy to the Slack Penalty Weight $\alpha$

Taking CartPole as a representative system, we analyze the sensitivity of QPKO's modeling accuracy to the penalty weight $\alpha$ in the QP-based mapping (9). Specifically, the NMAE of multi-step predictions given by QPKO is reported in Table 11. As shown in this table, the NMAE varies only slightly as $\alpha$ changes, indicating that the modeling accuracy of QPKO is robust to this hyperparameter.

*Table 11.* The NMAE given by QPKO on CartPole under different values of the penalty weight $\alpha$. Results are reported for prediction horizons $T \in \{50, 100, 150, 200\}$ as mean±std (in units of $10^{-2}$). The mean and standard deviation are computed over 5 independent training trials.

| $\alpha$ ($\times 10^{-2}$) | $T = 50$ | $T = 100$ | $T = 150$ | $T = 200$ |
|---|---|---|---|---|
| 0.6 | 0.324±0.053 | 0.666±0.098 | 1.068±0.161 | 1.767±0.237 |
| 0.8 | 0.326±0.050 | 0.666±0.094 | 1.087±0.164 | 1.811±0.246 |
| 1.0 | 0.325±0.057 | 0.663±0.096 | 1.073±0.165 | 1.783±0.257 |
| 1.2 | 0.328±0.056 | 0.672±0.103 | 1.098±0.179 | 1.829±0.289 |
| 1.4 | 0.333±0.054 | 0.668±0.094 | 1.078±0.174 | 1.796±0.277 |

## H. Additional Training Efficiency Results

### H.1. Per-Epoch Training Time of Each Framework

Table 12 reports the per-epoch training time of each framework. Since this time varies with the prediction horizon $T$, the presented values are averaged across all training horizons, i.e., $T \in \{10, 25, 40, 55, 70, 85\}$. According to this table, the per-epoch training time of QPKO is the shortest across the four systems. Together with the faster training convergence shown in Figure 3, this enables QPKO to achieve the shortest wall-clock training time, as shown in Table 2.

*Table 12.* Average per-epoch training time (in seconds) of each framework across four nonlinear dynamical systems on an NVIDIA RTX 4090 GPU. The best-performing results are highlighted in **bold**.

| FRAMEWORK | CARTPOLE | VDP | WT | TSR |
|---|---|---|---|---|
| DKO | 0.699 | 0.628 | 0.667 | 0.730 |
| EDMD-DL | 0.623 | 0.492 | 0.565 | 0.617 |
| NDMD | 0.663 | 0.562 | 0.706 | 0.755 |
| QPKO | **0.389** | **0.358** | **0.358** | **0.381** |

### H.2. Computational Cost of QPKO under Different Lifted Dimensions

Table 13 reports the per-iteration computation time and GPU memory usage of QPKO for different lifted dimensions $N$. As shown in this table, QPKO exhibits favorable scalability up to moderate dimensions, i.e., $N \leq 300$, on a single NVIDIA RTX 4090 GPU.

*Table 13.* Per-iteration computation time and GPU memory usage of QPKO for different lifted dimensions. These results are obtained on an NVIDIA RTX 4090 GPU with $T = 45$ and a batch size of 2048.

| $N$ | Time per Iteration (s) | Memory Usage (GB) |
|---|---|---|
| 100 | 0.18 | 0.75 |
| 200 | 1.04 | 1.83 |
| 300 | 3.30 | 4.83 |
| 400 | 7.30 | 10.15 |
| 500 | 14.00 | 19.27 |

