# OpenReview forum: "QPKO: Differentiable QP-Embedded Deep Koopman Framework for Modeling Nonlinear Systems"
_ICML.cc/2026/Conference — ICML 2026 regular_

### Official Review · Reviewer_j3hh · 2026-03-08

**Soundness:** 3
**Presentation:** 3
**Significance:** 2
**Originality:** 3
**Overall Recommendation:** 5
**Confidence:** 3

**Summary:**

The author presents a QP-based learning framework to approximate the dynamics of nonlinear systems. The paper is well-written, but some explanations and additional experiments are expected.

**Compliance With Llm Reviewing Policy:**

Affirmed.

**Final Justification:**

The paper is well-written and my concerns are addressed. To summarize, the soundness, significance, and clarity are high and the originality is moderate as the technologies used are developped before. However, in general, the paper is novel. The authors have addressed my concerns. In general, I suggest "5: Accept: Technically solid paper".

**Key Questions For Authors:**

1.	The motivation is a bit weak. In general, the claim is that end-to-end training is better. While it may be true, the author is suggested to provide more insights or experimental support.
2.	Fig. 1 as the framework is a bit messy. The author is suggested to remove some contents.
3.	What are x_1^{1:N_qp} and x_2^{1:N_qp}? Why can we have N_qp data here? Are they simply the cut transition tuple from the dynamical system data? The author should give an explanation.
4.	The author shoud explain the true data in Eq. 15. Are they computed by g(x)?
5.	Does the observable function g needs specific designs, e.g., activation functions? Otherwise, this is simply a feature processing and I can hardly find it’s deeper link to  Koopman operator.
6.	Can the author explain more why the method is better than treating A, B as independent trainable component? For example, we can directly remove QP, and treat A, B as trainable parameter matrices in the neural network. Then, we can minimize the one-step or multi-step mse loss. Will this be more efficient?
7.	Why does DKO requires longer training time? It seems that there is no QP inside it. Moreover, you may need to present the test time, as you define the training time quite differently (the time to reach the best validation performance) and test time is important for dynamic control.
8.	The author doesn’t discuss the potential drawback of the model.
9.	In fact, lifting original state to latent space can also be done using seq-to-seq models, such as neural ODE or latent ODE, although they never claimed a link to Koopman operator. The author is suggested to test them.

**Limitations:**

The author is suggested to discuss some limitations about robustness to noise, data scarcity, and more complex systems.

**Strengths And Weaknesses:**

The paper is well-written, and the theory is sound. The method seems to be the most accurate and efficient based on the results. However, some explanations and additional experiments are expected.

---

> ### Author Rebuttal · Authors · 2026-03-31
>
> We greatly appreciate your valuable and insightful comments. Please find our point-by-point reply below.
>
> ---
> **Reply to Q1, Q6, and Q7**
>
> In fact, our claim is that the QP-embedded end-to-end training is better. The rationale is as follows:
>
> * Compared with treating the global linear model (i.e., $\mathbf{A}$ and $\mathbf{B}$) as independent trainable components (DKO in our paper), our method has two advantages. (i) Instead of freely optimizing the observable functions and the global linear model in a highly non-convex space, it uses a QP problem to generate the QP-based optimal global linear model under the current learned observable functions, providing a basic guarantee on the modeling accuracy. (ii) Since the global linear model is not treated as an independent trainable component, the search space of the training is effectively reduced, which lowers the training complexity.
>
> * Compared with the alternating two-stage training (EDMD-DL in our paper), during observable function optimization, our method can account for the QP-based update of the global linear model via end-to-end gradient. In contrast, this update is ignored in the two-stage training, where the global linear is treated as a fixed component when optimizing the observable functions. This may degrade the training stability.
>
> The newly added experiments under noise scenarios further demonstrate the advantage of QP-embedded end-to-end training. (Please see “Q6” in our response to Reviewer SGip.)
>
> The test time (in milliseconds) of the model given by each framework at $T=200$ is reported in the table below.
>
> |Framework|CartPole|VDP|WT|TSR|
> |:---|:---|:---|:---|:---|
> |**DKO**|26.85|26.35|26.58|26.96|
> |**EDMD-DL**|26.55|26.25|26.26|26.94|
> |**NDMD**|26.73|26.58|27.07|27.18|
> |**QPKO**|26.69|26.57|26.33|27.16|
>
> This table shows that the model given by each framework exhibits similar test time. This is because, although these frameworks differ in their training strategies, the learned models share the same structure, as detailed in Equation (3).
>
> ---
> **Reply to Q2**
>
> We will revise Figure 1 to make it clearer and more concise.
>
> ---
> **Reply to Q3**
>
> Yes, $\mathbf{x}_ 1^{1:N_{\text{QP}}}$ and $\mathbf{x}_ 2^{1:N_{\text{QP}}}$ are obtained by reorganizing the trajectory data of the system into one-step transition tuples. Specifically, the former denotes the first states in the $ N_{\text{QP}}$ tuples, and the latter denotes the corresponding successor states. These tuples together constitute the QP dataset, which is used to compute the global linear model via the differentiable QP layer. Using $N_{\text{QP}}$ tuples, rather than a single tuple, is important for computing a **global** linear model.
>
> ---
> **Reply to Q4**
>
> Yes, the true data in (15) is computed by $g$, i.e., $\mathbf{z}_ {k+1}=g(\mathbf{x}_{k+1};\boldsymbol{\theta})$.
>
> ---
> **Reply to Q5**
>
> In common, the architecture of the NN in $g$ does not require a highly specific design. This is consistent with many existing studies in the deep Koopman framework, where a standard MLP is commonly used as a universal approximator to learn finite-dimensional observable functions or eigenfunction representations of the Koopman operator.
>
> From a purely deep learning perspective, $g$ can indeed be viewed as a feature-processing block. However, Koopman operator theory gives this feature processing a specific meaning. That is, the learned features are required to serve as observable coordinates in a high-dimensional lifted space, where the dynamics of the original nonlinear system can be expressed in a linear manner, i.e., the formulation in equation (3). We will further clarify this link in the revised paper.
>
> ---
> **Reply to Q8 and Limitation**
>
> (i) According to time per iteration and memory usage in different lifted dimensions $N$, QPKO is currently applicable to moderate dimensions, i.e., $N≤300$ (Please see “W.4 and Q5” in our response to Reviewer SGip). (ii) As a purely data-driven framework, the modeling accuracy of QPKO is closely related to the quantity and quality of the training data. Although QPKO outperforms all baselines under noise scenarios, its accuracy degradation is still relatively noticeable (please see “Q6” in our response to Reviewer SGip). Therefore, more effort is still needed to improve the scalability of QPKO and reduce its data reliance.
>
> ---
> **Reply to Q9**
>
> Models such as neural ODE or latent ODE are indeed effective for predicting nonlinear dynamics. However, there is a fundamental difference between these models and those derived from Koopman-based modeling: the former are inherently nonlinear models, whereas the latter are global linear models.
>
> The most critical advantage of Koopman modeling is exactly this linearity. It allows well-established control (such as MPC) and analysis techniques for linear systems to be applied to nonlinear systems. In this sense, although neural ODE or latent ODE are, to some extent, related, they are outside of the core scope of the current paper.

---

> > ### Author Rebuttal · Reviewer_j3hh · 2026-04-01
> >
> > Thanks for your responses. I will consider raising the score.

---

> > > ### Author Response · Authors · 2026-04-01
> > >
> > > We are really excited and honored that our responses have adequately addressed your concerns. We deeply appreciate the time and effort you have invested in providing us with such valuable and insightful feedback, which is really important and helpful for our research.

---

### Official Review · Reviewer_SGip · 2026-03-12

**Soundness:** 3
**Presentation:** 3
**Significance:** 3
**Originality:** 3
**Overall Recommendation:** 5
**Confidence:** 5

**Summary:**

The paper presents QPKO, a novel algorithm that addresses key limitations of existing methods in the deep learning Koopman framework. It focuses on improving both model accuracy and training efficiency. Similar to previous approaches, the observable functions are parameterized; however, the core innovation lies in mapping these observables to a linear (lifted) model through a quadratic programming (QP) problem, which significantly reduces optimization complexity. The algorithm is evaluated on several examples, demonstrating clear superiority over other methods.

**Compliance With Llm Reviewing Policy:**

Affirmed.

**Final Justification:**

I believe that this was a strong paper to begin with. The reviewers have addressed my comments in large, with the exception being my question regarding eigendecompositions of the learned models. However, given that this is not the central concern in the control setting, I am keeping my positive score.

**Key Questions For Authors:**

- Slack variables are usually non-negative. Is this true for your version as well? If yes/no, then why? Does this lead to pathological solutions or numerical issues?
- Would it be feasible to directly incorporate a multi-step prediction loss into the objective? (In part answering my own question, I guess that this only works for the non-controlled case, though)
- can you elaborate some more on the different training sets $D_{QP}$ and $D_{Tra}$?
- How sensitive are the results to the penalty $\alpha$ and the margin $c$ in the QP?
- How does QPKO scale with the lifted dimension $N$ and dataset size $N_{QP}$? Have you assessed memory usage and runtime for $N$ in the several hundreds or thousands?
- Have you evaluated robustness to noise? How do the stability constraints interact with noisy transitions?
- In Eq. (2), you wrap the lifting $\psi$ around both $x$ and $u$, which is quite uncommon and impractical. Maybe you could change this, since you don't lift the control in the end anyway?

**Limitations:**

A discussion of the limitations would be very much appreciated. For instance in terms of
- scaling towards larger dimensions
- sensitivity in the presence of noise

**Strengths And Weaknesses:**

**Strengths**
- The idea of replacing the training of A and B with a differentiable, constrained QP that couples one-step fitting with multi-step-oriented stability constraints is interesting and innovative.
- The overall pipeline is clearly explained.
- The authors provide rigorous proofs (Theorem 4.1 and Proposition 4.2) to validate the effectiveness of their improvements.
- They demonstrate the algorithm’s performance on numerous examples and compare it with other three existing methods, highlighting enhancements in accuracy, efficiency, and control performance.

**Weaknesses**
- Besides a linear model, the Koopman business is often about eigenfunctions and stability analysis of the Koopman operator (or the derived approximation $A$). Have you studied these Eigenfunctions for some of your examples? Vor van der Pol, there are many papers who study this, for instance.
- the practical realization of Eq. (13) is not very clear
- a discussion of current limitations and shortcomings would be appreciated
- the experiments are limited to rather low-dimensional examples. The real challenges often only occur for larger dimensions, for instance when discretizing PDEs. I recommend that the authors include a discussion on larger-dimensional systems
- discussion of the hyperparameters could be improved

---

> ### Author Rebuttal · Authors · 2026-03-31
>
> We deeply appreciate your insightful and valuable comments.
>
> ---
> **W.1**
>
> To reveal the dynamical behavior of nonlinear systems, many studies were conducted to analyze eigenfunctions and spectral properties of the Koopman operator. However, such analysis is mainly conducted for uncontrolled systems, rather than controlled systems, whose behavior depends not only on the system itself but also strongly on external inputs.
>
> Although QPKO is also applicable to uncontrolled systems, this paper focuses on controlled systems. Accordingly, the goal of the Koopman approximation here is to provide a predictive model for downstream model-based control rather than to support spectral analysis. Hence, in subsection 5.3, we evaluate the learned model in MPC. Inspired by this comment, leveraging QPKO for eigenfunction and spectral analysis will be an important direction for our future work.
>
> ---
> **W.4 and Q5**
>
> According to the experimental result, both run time and memory usage are highly sensitive to $N$ but much less sensitive to $N_\text{QP}$. The table below reports these metrics for varying $N$ (on an RTX 4090 GPU at $T=45$ and a batch size of 2048).
>
> |$N$|Time per Iteration (s)|Memory Usage (GB)|
> |:---|:---:|:---: |
> |**100**|0.18|0.75|
> |**200**|1.04|1.83|
> |**300**|3.30|4.83|
> |**400**|7.30|10.15|
> |**500**|14.00|19.27|
>
> This table shows that QPKO is applicable for moderate dimensions, i.e., $N≤300$. Although QP decomposition improves scalability, the cost is still high when $N>300$. In the future, more effort will be devoted to further improving the scalability.
>
> Nevertheless, the current scalability of QPKO is sufficient for many ODE-based controlled systems. For much higher-dimensional systems, such as discretized PDEs, a potential way to apply QPKO is dimensionality reduction. This way has been demonstrated effective in many studies. For example, in [R1], the spatial dimension of a turbulence system is 295,122, while the Koopman-based analysis is conducted on 250 observables.
>
> [R1] Xu et al., ResKoopNet: Learning Koopman Representations for Complex Dynamics with Spectral Residuals. ICML 2025.
>
> ---
> **W.5 and Q4**
>
> In QPKO, $c$ is not a tunable hyperparameter. It is only a small constant used to replace $≤$ with $<$. On CartPole, the table below (a simplified version) reports the sensitivity of the modeling accuracy to $\alpha$.
>
> |$\alpha (\times10^{-3})$ \ $T$|100|200|
> |:---|:---:|:---:|
> |**6**|0.666|1.767|
> |**8**|0.666|1.811|
> |**10**|0.663|1.783|
> |**12**|0.672|1.829|
> |**14**|0.668|1.796|
>
> These results show that QPKO is robust to the choice of $\alpha$.
>
> ---
> **Q1**
>
> In fact, in QPKO, $\xi$ is non-positive, since only in this case can it relax the constraint. We did not impose $\xi≤0$, because any positive $\xi$ would both tighten the constraint and increase the slack penalty, and thus would never be chosen by the optimizer. Meanwhile, this formulation does not cause pathological solutions or numerical issues, as (9) remains a standard QP.
>
> ---
> **Q2**
>
> Even in the non-controlled case, directly incorporating a multi-step loss would make the objective non-quadratic. Hence, the corresponding optimization would no longer be a QP, but a non-convex problem.
>
> ---
> **Q6**
>
> On CartPole, the tables below (a simplified version) report the modeling accuracy under two noise levels, i.e., 3% and 6%. The results show that QPKO achieves higher modeling accuracy at both noise levels.
>
> * 3%
> |Framework \ $T$|100|200|
> |:---|:---|:---|
> |**DKO**|1.114|3.431|
> |**EDMD-DL**|1.239|3.520|
> |**NDMD**|0.930|2.868|
> |**QPKO**|**0.716**|**2.107**|
>
> * 6%
> |Framework \ $T$|100|200|
> |:---|:---|:---|
> |**DKO**|1.156|3.503|
> |**EDMD-DL**|2.021|4.921|
> |**NDMD**|1.116|3.407|
> |**QPKO**|**0.753**|**2.268**|
>
> We also conduct an ablation study on constraint (8) under each noise level, as reported in the tables below (a simplified version).
>
> * 3%
> |Framework \ $T$|100|200|
> |:---|:---|:---|
> |**w/o (8)**|0.724|2.296|
> |**w (8)**|**0.716**|**2.107**|
>
> * 6%
> |Framework \ $T$|100|200|
> |:---|:---|:---|
> |**w/o (8)**|0.801|2.822|
> |**w (8)**|**0.753**|**2.268**|
>
> According to the above two tables and Table 8, the benefits given by constraint (8) are more obvious under the higher-noise (6%) scenario than in the noise-free and low-noise (3%) scenarios.
>
> ---
> **Q7**
>
> Your observation is correct. The formulation (2) corresponds to the rigorous extension of Koopman operator theory to controlled systems, but it is not suitable for practical implementation. To address this issue, [R2] reformulated it into a more practical and implementable form, i.e., (3).
>
> [R2] Korda et al., Linear predictors for nonlinear dynamical systems: Koopman operator meets model predictive control. Automatica 2018.
>
> ---
> **W.2, W.3, Limitation 1, and Q3**
>
> Thanks for these comments. We will further clarify the practical realization of Eq. (13) and elaborate on the two datasets in the revised paper. Please find our discussion on limitations under “Reply to Q8 and Limitation” in our response to Reviewer j3hh.

---

> > ### Author Rebuttal · Reviewer_SGip · 2026-04-02
> >
> > Thank you for the clarifications, even though I disagree a little bit with your reply to W1. There are, in fact, results on extracting eigendecomositions from actuated data [1]. If you have identified the correct model, then setting the control to zero would yield the dynamics of the non-actuated system.
> >
> > Nevertheless, I am satisfied with the reviewers replies and it's ok for me to leave this topic to future work. Given the fact that I already voted "5: Accept", I am not changing my score.
> >
> > [1] Otto et al. Learning Bilinear Models of Actuated Koopman Generators from Partially Observed Trajectories. SIADS 23(1), 2024. DOI: 10.1137/22M1523601

---

> > > ### Author Response · Authors · 2026-04-02
> > >
> > > Thank you very much for your positive and enlightening feedback. We especially appreciate you pointing us to the literature [1]. We will study this work carefully and leverage its valuable insights to further improve our future work.

---

### Official Review · Reviewer_zCEF · 2026-03-12

**Soundness:** 3
**Presentation:** 3
**Significance:** 3
**Originality:** 3
**Overall Recommendation:** 4
**Confidence:** 4

**Summary:**

This paper proposes a new method for learning finite-dimensional approximations of the Koopman operator for nonlinear dynamical systems using deep learning. The authors introduce a framework called QPKO, which embeds a quadratic program (QP) within the training pipeline to determine the global linear model associated with the learned observable functions. The QP objective is designed to minimize one-step prediction error while incorporating constraints intended to improve multi-step (effectively long-horizon) prediction accuracy by enforcing stability properties of the learned linear dynamics. The resulting QP-based mapping from observable functions to the linear model is implemented as a differentiable layer using OptNet, enabling end-to-end training. To improve computational efficiency, the authors further decompose the large QP into multiple smaller sub-problems that can be solved in parallel on GPU. Experiments on several nonlinear dynamical systems demonstrate improvements over baseline deep Koopman frameworks in terms of prediction accuracy (measured by NMAE), training efficiency, and downstream control performance.

**Compliance With Llm Reviewing Policy:**

Affirmed.

**Key Questions For Authors:**

[Q1] Could the authors clarify the practical motivation behind enforcing $||A||_{\infty} < 1$ through the QP constraint? In particular, how does this constraint contribute to improving multi-step prediction accuracy beyond ensuring bounded cumulative error?

[Q2] Why was the $\ell_\infty$​ norm chosen in Theorem 4.1? Would alternative formulations such as an $\ell_2$​-based constraint (e.g., $||A||_2 < 1$) lead to different modeling or stability properties? I think this should lead to an SDP based problem formulation.

[Q3] Since the authors impose a constraint that $||A||_\infty < 1$, could the authors comment on how this method would behave when it is trained for marginally stable nonlinear dynamical systems?

[Q3] For the TSR experiments, the control inputs are generated from a predetermined control law. How well does the learned model generalize to control sequences that differ from this policy?

[Q4] Could the authors provide additional detail on the training efficiency comparison? In particular, it would be helpful to report per-epoch training time for the different methods to better understand the source of the wall-clock improvements.

**Limitations:**

Yes.

**Strengths And Weaknesses:**

Strengths

[S.1] The technical development in the paper appears largely sound. In particular, the theoretical statements and accompanying proofs (e.g., Theorem 4.1 and related derivations) appear correct and consistent with the assumptions made in the paper.

[S.2] The paper proposes a novel framework that integrates a quadratic programming (QP) formulation with deep Koopman learning. By leveraging OptNet, the authors embed the QP as a differentiable layer, enabling end-to-end training. Additionally, the proposed decomposition of the large QP into multiple smaller sub-QPs allows the method to be solved efficiently in parallel on GPUs.

[S.3] The empirical evaluation demonstrates consistent improvements over the considered baselines in terms of prediction accuracy (NMAE) and training efficiency across multiple benchmark nonlinear dynamical systems.

[S.4] The paper is clearly written and generally easy to follow. The motivation for the method, the architectural design, and the training pipeline are well explained.

Weaknesses

[W.1] The motivation behind Theorem 4.1 is somewhat unclear. The theorem shows that if $||A||_{\infty}<1$, then the cumulative prediction error is bounded. However, boundedness does not necessarily imply that the cumulative prediction error is minimized. Since this condition is introduced as a constraint in the QP formulation, its direct impact on improving long-horizon prediction performance is not fully justified.

[W.2] The choice of the $\ell_\infty$​ norm in Theorem 4.1 is not well motivated. The bounded-error argument would hold for other matrix norms as well, and the paper does not clearly explain why this particular norm is preferable.

[W.3] The constraint $||A||_\infty < 1$ enforces strict stability of the learned linear dynamics. While this may help control prediction error growth, it may also restrict the expressiveness of the learned Koopman approximation, especially for systems with marginally stable or unstable modes. It would be helpful for the authors to discuss whether this constraint could limit the ability of the model to represent certain classes of dynamical systems.

[W.4] In the TSR dataset, the control inputs are generated using a predetermined control law. This raises concerns about whether the learned model generalizes to arbitrary control sequences outside the distribution induced by this control policy.

[W.5] The wall-clock training time comparison lacks clarity regarding the source of the improvement. It is unclear whether the reported speedups arise from lower per-epoch computation time or from faster convergence.

---

> ### Author Rebuttal · Authors · 2026-03-31
>
> **Reply to W.1 and Q1**
>
> Thanks for these comments. In fact, the main motivation for incorporating this constraint is not to directly reduce the prediction error, but rather to **better align the objectives of the inner and outer optimizations in QPKO**.
>
> Specifically, if without constraint (8), the inner optimization within the differentiable QP layer would focus only on minimizing the one-step prediction error. However, the outer optimization is to reduce the multi-step prediction loss in (15). This leads to an objective mismatch problem.
>
> The constraint (8) is precisely designed to mitigate this problem. Although it cannot directly embed the minimization of multi-step error into the inner optimization’s objective, it works as a regularizer that biases the differentiable QP layer toward generating a global linear model that is less likely to amplify rollout errors over long prediction horizons. In this sense, it allows the inner optimization to account for multi-step prediction behavior, thereby mitigating the objective mismatch.
>
> The ablation studies in both Appendix F and “Q6” of our response to Reviewer SGip demonstrate the effectiveness of this constraint.
>
> ---
> **Reply to W.2 and Q2**
>
> Thanks for these comments. The choice of the infinity norm is further guided by two additional principles:
>
> * **The resulting norm constraint can be equivalently transformed into linear constraints.** This is to preserve a QP formulation for the differentiable optimization layer. Compared with the SDP formulation, the QP formulation is typically more computationally tractable.
>
> * **The resulting QP with these transformed linear constraints can be equivalently decomposed into $N$ independent sub-QP problems.** As the OptNet can solve and back-propagate the obtained sub-QP problems in parallel on GPU. This principle improves the scalability of QPKO.
>
> If only the first additional principle is considered, both the 1-norm and the infinity norm are viable choices. However, the 1-norm does not satisfy the second principle, whereas the infinity norm does.
>
> ---
> **Reply to W.3 and Q3(1)**
>
> Thanks for these comments. For the nonlinear system, whose lifted global linear model is inherently marginally stable or unstable, strictly enforcing constraint (8) could indeed restrict the learning of the Koopman approximation.
>
> However, **in QPKO, the constraint (8) is not incorporated as a hard constraint**. Instead, as shown in (9), we soften this strict condition by introducing slack variables $\xi_{i}$, which are penalized in the objective function. This slacked formulation allows a controlled violation of the constraint when necessary to accurately capture marginally stable or unstable dynamics.
>
> Moreover, the objective of the outer optimization is to reduce the multi-step rollout loss (15). If the inner optimization within the differentiable QP layer exhibits an excessively stable bias that harms this objective, the end-to-end gradients can adjust the observable functions to steer the generated global linear model away from such a bias.
>
> Nevertheless, we have to acknowledge that the slacked constraint may still affect the optimality of the learned model in largely unstable scenarios. We will clarify this potential limitation in the revised paper.
>
> ---
> **Reply to W.4 and Q3(2)**
>
> Thanks for these important comments regarding the model’s generalization ability w.r.t. control policies.
>
> One important purpose of subsection 5.3 is precisely to verify this ability. Taking TSR as an example, the QPKO is trained on a dataset collected under a predetermined control policy, i.e., $u_ {k} = 4x_ {3,k} + 2x_ {4,k} + 3$. The resulting global linear model is then evaluated under a completely different control policy, i.e., MPC. The predetermined control policy and MPC differ substantially and belong to fundamentally different control paradigms: the former is an explicit control policy, while the latter is a model-based optimal control policy.
>
> According to Table 3 and Figure 4(d), across 50 control trials, the MPC controller based on the model learned by QPKO achieves ideal control performance on TSR, while the MPC controllers based on the three baselines all fail to accomplish the control task. This result indicates that QPKO exhibits satisfactory generalization ability under the control policy shift.
>
> ---
> **Reply to W.5 and Q4**
>
> Thanks for these valuable comments. The per-epoch training time of each framework is reported in the table below (in seconds). As this time is impacted by $T$, the reported values are averaged over all $T$ used during training.
>
> | Framework | CartPole | VDP | WT | TSR |
> | :--- | :--- | :--- | :--- | :--- |
> | DKO | 0.699 | 0.628 | 0.667 | 0.730 |
> | EDMD-DL | 0.623 | 0.492 | 0.565 | 0.617 |
> | NDMD | 0.663 | 0.562 | 0.706 | 0.755 |
> | QPKO | **0.389** | **0.358** | **0.358** | **0.381** |
>
> This table and Figure 3 indicate that **the lower training time comes from both lower per-epoch training time and faster convergence**.

---

> > ### Author Rebuttal · Reviewer_zCEF · 2026-04-07
> >
> > I thank the authors for the detailed and thoughtful responses. The rebuttal clarifies several of my concerns.
> >
> > Regarding W.1, the explanation that constraint (8) is intended to mitigate the objective mismatch between one-step prediction (inner QP) and multi-step rollout loss (outer optimization) is helpful. Framing the constraint as a regularizer that biases the learned linear model toward better long-horizon behavior provides a clearer motivation.
> >
> > For W.2, the discussion on the choice of the
> > ℓ
> > ∞
> > ℓ
> > ∞
> > 	​
> >
> >  norm is convincing. In particular, the arguments about preserving a QP formulation and enabling decomposition into independent sub-QPs for efficient parallelization clarify the practical design trade-offs compared to alternatives such as SDP formulations.
> >
> > For W.3, I appreciate the clarification that the stability constraint is implemented as a soft constraint via slack variables. This alleviates concerns about overly restricting expressiveness, although the authors appropriately acknowledge that this may still affect performance in highly unstable regimes.
> >
> > Regarding W.4, the additional explanation of the TSR experiment and the evaluation under a different control policy (MPC) is helpful and provides evidence that the learned model generalizes beyond the training distribution.
> >
> > Finally, for W.5, the inclusion of per-epoch training time resolves the ambiguity in the reported efficiency improvements.
> >
> > Overall, the rebuttal strengthens the paper by clarifying both the motivation and practical implementation details. My overall assessment remains unchanged.

---

> > > ### Author Response · Authors · 2026-04-07
> > >
> > > Thanks for your positive and valuable feedback. We are really happy to hear that our response has successfully clarified your concerns and strengthened the paper. If there is anything else we could further clarify and elaborate on, we are fully available to address it.

---

### Official Review · Reviewer_BuiY · 2026-03-13

**Soundness:** 3
**Presentation:** 4
**Significance:** 2
**Originality:** 2
**Overall Recommendation:** 5
**Confidence:** 4

**Summary:**

The paper considers the problem of learning nonlinear dynamical system models with inputs. It is based on the Koopman theory which maps  a nonlinear system to a linear one and then learning the nonlinear mapping and the linear dynamics. The paper proposes a new training methodolgy, including an appropriately designed loss function where gradients can be computed. The results are supported with numerical experiments.

**Compliance With Llm Reviewing Policy:**

Affirmed.

**Final Justification:**

A detailed rebuttal, discussed further the limitations I pointed, if these limitations are clearly stated in the final version, this improves my assessment

**Key Questions For Authors:**

- it is not clear if adding the constrain that the norm of A must be less than one allows learning A that are unstable, for unstable systems that need to be stabilized.

- There is something confusing. Here the QP layer depends on a bunch of data. In general Neural nets with QP layers, as far as I can tell, there are no data points in the definition of the QP problem, but then ones defines a loss function for the neural net with the QP layer, where the loss function is an average over data. So the present paper has a different way of thinking about the QP layer and this is in contrast to the literature.

- Can this be pushed in more complicated (like higher state dimension) problems? Why yes or why not?

**Limitations:**

- The ablation study in section F is appreciated, I had the same question. But the results seem to know that many times there is only a small margin of improvement with the more complex constraints. Also, without the constraints, wouldn't the algorithm be faster?

- The QP problem is defined based on data. Are these data the same as the data used in the rollout trajectories in (15)?

- For clarity, I think you should include a minimization problem where you minimize (15) over theta

-

**Strengths And Weaknesses:**

- The paper is very clear to read.

- The training methodology, based on QPs and QP layers, is novel compared to the literature. It also allows end to end training

- The numerical results show clearly the improved accuracy and efficiency of the training. The addition of control schemes in MPC is very helpful for control applications.

- Some parts in the problem formulation are unclear. Theorem 4.1 says that if the matrix norm of A is less than one then the long term prediction error is bounded. This statement seems too loose to be useful. Then the authors proceed by adding the condition on A as a constraint. It is unclear what this constraint achieves, as the theorem does not say that the prediction error decreases when that norm is decreasing for example.

---

> ### Author Rebuttal · Authors · 2026-03-31
>
> We deeply appreciate all your valuable comments.
>
> ---
> **Reply to "4th comment in Strengths And Weaknesses" and Q1**
>
> Thanks for these valuable comments on the motivation behind the constraint (8) that is formulated based on the condition $\left\lVert \mathbf{A}\right\rVert_\infty<1$, as well as its potential impact on the learning of $\mathbf{A}$.
>
> Reviewer zCEF raised very similar concerns. Due to the 5,000-character limit, we kindly direct you to our **response to Reviewer zCEF**, where we provide a detailed discussion on:
>
> * The motivation behind constraint (8) (please see **Reply to W.1 and Q1**).
> * The potential impact of this constraint on the learning of $\\mathbf{A}$ (please see **Reply to W.3 and Q3(1)**).
>
> ---
> **Reply to Q2**
>
> Yes, in QPKO, the differentiable QP layer is indeed leveraged in a different manner.
>
> In the conventional setting, the QP parameters within the QP layer are determined by a single data point. That is, given an input $\mathbf{x}^i$, the NNs output $\mathbf{y}_ {\text{NN}}^i$. Then, based on $\mathbf{y}_{\text{NN}}^i$, the QP parameters are determined, and the corresponding QP solution $\mathbf{s}^*$ is computed. This solution is used to compute the loss for the given input. A batch of inputs results in a batch of losses, which are averaged and backpropagated.
>
> In contrast, in QPKO, the QP parameters are determined by "a bunch of" one-step transition tuples collected from the modeled nonlinear system, i.e., $\{\mathbf{x}_ 1^{1:N_{\text{QP}}}, \mathbf{x}_ 2^{1:N_{\text{QP}}}, \mathbf{u}_ 1^{1:N_{\text{QP}}}\}$. The reason is that the differentiable QP layer within QPKO is developed to compute a global linear model (i.e., $\mathbf{A}$ and $\mathbf{B}$) corresponding to the currently learned observable functions. To ensure that the computed model captures the global linear dynamics in the lifted space, rather than merely a sample-wise local mapping, the QP parameters should be determined from a collection of transition tuples rather than from a single data point.
>
> ---
> **Reply to Q3**
>
> Yes, QPKO holds potential for more complicated problems. This can be attributed to our specifically designed OptNet-based differentiable QP layer. The rationale is as follows:
>
> Firstly, we interpret “more complicated problems” primarily as higher-dimensional problems, since the computational complexity of QPKO is mainly affected by the lifted state dimension $N$. This effect is mainly reflected in the complexity of the QP problem (9). Since this problem consists of $2N^2+Nm+N$ decision variables and $2N^2+N$ constraints, its complexity grows rapidly as $N$ increases, hindering the scalability of QPKO to high-dimensional applications.
>
> This issue is effectively mitigated by our specifically designed OptNet-based differentiable QP layer. Specifically, in this layer, the original QP problem is decomposed into $N$ independent sub-QP problems. The complexity of the subproblem grows much more slowly w.r.t. $N$, as it contains only $2N+m+1$ decision variables and $2N+1$ constraints. Furthermore, by leveraging OptNet, these subproblems are efficiently solved and back-propagated in parallel on GPUs. These two factors effectively improve the scalability of QPKO to higher-dimensional cases.
>
> However, the potential of QPKO for more complicated problems is not unlimited. We detail this limitation under “W.4 and Q5” in our response to Reviewer SGip.
>
> ---
> **Reply to Limitation 1**
>
> We fully understand your concern about the trade-off between accuracy and efficiency. Our rationale for retaining constraint (8) is threefold:
>
> * The main motivation of introducing constraint (8) is to better align the inner and outer optimization objectives, thereby making it more likely for QPKO to achieve higher modeling accuracy. This motivation is detailed under “Reply to W.1 and Q1” in our response to Reviewer zCEF.
>
> * According to Table 8, this motivation indeed leads to an overall improvement in accuracy. Although this improvement is not always obvious, in some cases the improvement is significant. Specifically, on VDP at $T$ = 100, 150, and 200, the error reduction ratios are 41.59%, 39.09%, and 28.88%, respectively. On TSR at $T$ = 50, this ratio is 36.61%. Moreover, in the higher noise level (6%), this constraint obviously improves the modeling accuracy (please see “Q6” in our response to Reviewer SGip).
>
> * QPKO already exhibits the advantage in modeling efficiency. This makes it, to some extent, not worthwhile to trade a potential gain in accuracy for additional efficiency.
>
> ---
> **Reply to Limitation 2**
>
> Yes. Both the data used to define the QP problem (i.e., $\mathcal{D}_ {\text{QP}}$) and the data used to compute the loss (i.e., $\mathcal{D}_ {\text{Tra}}$) in (15) are constructed from trajectories in the training set. The difference lies only in how these trajectories are reorganized.
>
> ---
> **Reply to Limitation 3**
>
> Thanks for this constructive suggestion. We will include such a minimization problem in our revised paper.

---

> > ### Author Rebuttal · Reviewer_BuiY · 2026-04-03
> >
> > I think the authors have not responded to my question :  the theorem does not say that the prediction error decreases when that norm is decreasing

---

> > > ### Author Response · Authors · 2026-04-04
> > >
> > > Thanks for your response. We sincerely appreciate your rigorous and insightful comments again, and we also apologize that our previous reply did not effectively address your question, i.e., “Theorem 4.1 does not say that the prediction error decreases when $\lVert\mathbf{A}\rVert_{\infty}$ is decreasing”.
> > >
> > > Your opinion is absolutely right. Theorem 4.1 indeed only indicates that when $\lVert\mathbf{A}\rVert_{\infty} < 1$, the long-term prediction error is bounded. It does not state that the prediction error decreases as $\lVert\mathbf{A}\rVert_{\infty}$ decreases. More precisely, the statement that “the prediction error decreases monotonically as $\lVert\mathbf{A}\rVert_{\infty}$ decreases” is not guaranteed in general.
> > >
> > > Precisely for this reason, we have not attempted to introduce some mechanisms in the differentiable QP layer, such as replacing the first constraint in the QP problem (9) (i.e., $\sum_{j=1}^{N} r_{i,j}+\xi_{i} \le \mathbf{1}-c, i=1,\ldots,N$) with $\sum_{j=1}^{N} r_{i,j}+\xi_{i} \le \mathbf{0}-c, i=1,\ldots,N$, to decrease $\lVert\mathbf{A}\rVert_{\infty}$ as much as possible during the optimization process.
> > >
> > > In contrast, we leverage the property that the long-term prediction error is bounded when $\lVert\mathbf{A}\rVert_{\infty} < 1$ to formulate constraint (8) and incorporate its relaxed version into the QP problem (9). As the reviewer correctly pointed out, according to Theorem 4.1, this formulation indeed cannot make the QP problem minimize the long-term prediction error in a direct manner, and this is also not our intention in designing such a formulation.
> > >
> > > The motivation behind incorporating the relaxed constraint (8) into the QP problem (9) is to better align the objectives of the inner and outer optimizations within QPKO. Specifically, without this constraint, the inner optimization, i.e., the QP problem (9), will focus only on the one-step prediction and ignore the multi-step prediction. However, the outer optimization is to reduce the multi-step prediction loss in (15). This will lead to an objective mismatch between the inner and outer optimizations.
> > >
> > > The relaxed constraint (8) is designed to mitigate this problem. By penalizing the violation of the condition $\lVert\mathbf{A}\rVert_{\infty} < 1$, it regularizes the QP problem (9) toward generating a global linear model that is less likely to amplify rollout errors over long-term prediction. By this means, although it does not directly minimize the long-term prediction error, it allows the inner optimization, to some extent, to account for multi-step prediction rather than focusing only on one-step prediction.
> > >
> > > Inspired by your valuable and rigorous comment, we fully agree that the current design is not a complete solution to the aforementioned objective mismatch problem. A better solution would ideally (i) enable the inner optimization to directly minimize the long-term prediction error, (ii) ensure that the inner optimization remains a tractable QP problem, and (iii) allow this tractable QP problem to be effectively decomposed to improve scalability. However, finding such a solution is challenging. This is because directly incorporating the multi-step prediction error into the objective will make it non-quadratic. Hence, the corresponding optimization is no longer a QP problem, but an intractable non-convex optimization problem. We will clarify this limitation in the revised paper and list finding such a solution as an important direction in our future research.
> > >
> > > Nevertheless, the current design is empirically effective. Specifically, according to the ablation study under noise-free scenarios reported in Table 8, this design consistently improves the modeling accuracy of QPKO across prediction horizons and nonlinear systems. For example, on VDP at $T$ = 100, 150, and 200, the error reduction ratios reach 41.59%, 39.09%, and 28.88%, respectively. On TSR at $T$ = 50, this ratio is 36.61%. Moreover, the results of the newly added ablation study on CartPole under two noise levels (i.e., 3% and 6%) are reported in the tables below.
> > >
> > > * 3%
> > > |Framework \ $T$|50|100|150|200|
> > > |:---|:---|:---|:---|:---|
> > > |**w/o (8)**|0.357±0.051|0.724±0.093|1.299±0.192|2.296±0.341|
> > > |**w (8)**|**0.354±0.048**|**0.716±0.077**|**1.220±0.160**|**2.107±0.271**|
> > >
> > > * 6%
> > > |Framework \ $T$|50|100|150|200|
> > > |:---|:---|:---|:---|:---|
> > > |**w/o (8)**|0.408±0.067|0.801±0.113|1.539±0.277|2.822±0.530|
> > > |**w (8)**|**0.391±0.055**|**0.753±0.075**|**1.313±0.149**|**2.268±0.319**|
> > >
> > > According to these tables, the relaxed constraint (8) consistently improves the modeling accuracy of QPKO under noise scenarios as well, especially at the higher noise level (6%).
> > >
> > > Thank you again for this rigorous and insightful comment. It is really helpful not only for improving the current paper, but also for guiding our future work.

---

### Decision · Program_Chairs · 2026-04-30

**Decision:**

Accept (regular)

**Comment:**

This submission proposes QPKO, a differentiable QP-embedded deep Koopman framework for learning nonlinear dynamical systems. Reviewers agreed that the paper is clearly written, technically solid, and presents a novel integration of a QP layer into Koopman learning that reduces optimization complexity while enabling end-to-end training. The empirical results were also viewed positively, with consistent gains over relevant baselines in prediction accuracy, training efficiency, and downstream control performance. The author responses were helpful and resolved most of the major concerns. In particular, the rebuttal clarified the role of the stability-related constraint as a regularizer that mitigates the mismatch between the inner one-step optimization and the outer multi-step training objective, justified the choice of the infinity norm from the standpoint of preserving a tractable and decomposable QP formulation, and added useful discussion on per-epoch efficiency, robustness to noise, dataset construction, and scalability with the lifted dimension. These clarifications strengthened the paper, and several reviewers indicated that their concerns were adequately addressed or substantially clarified. At the same time, some reservations remain. In particular, the theoretical justification for the constraint is still somewhat indirect, since the boundedness argument does not directly imply improved long-horizon prediction, and the experimental study remains limited to relatively low-dimensional systems. These issues do not undermine the technical contribution, but they do limit the breadth of the paper’s impact and leave some questions about scalability and generality for future work. Overall, I view this as a technically sound and interesting contribution that advances the literature on deep Koopman modeling, with a meaningful methodological novelty and solid empirical support. Given the remaining limitations, my recommendation is Accept if there is space.

PC: weak accept